
# Multi-timescale variations of modelled stratospheric water vapor derived from three modern reanalysis products

Mengchu Tao[1], Paul Konopka[1], Felix Ploeger[1,2], Xiaolu Yan[1], Jonathon S. Wright[3], Mohamadou Diallo[1], Stephan Fueglistaler[4], and Martin Riese[1]

[1]Forschungszentrum Jülich (IEK-7: Stratosphere), Jülich, Germany
[2]Department of Physics, University of Wuppertal, Wuppertal, Germany
[3]Department of Earth System Science, Tsinghua University, Beijing, China
[4]Princeton University, Princeton, USA

**Correspondence:** Mengchu Tao (m.tao@fz-juelich.de)

**Abstract.** Stratospheric water vapor (SWV) plays important roles in the radiation budget and ozone chemistry and is a valuable tracer for understanding stratospheric transport. Meteorological reanalyses provide variables necessary for simulating this transport; however, even recent reanalyses are subject to substantial uncertainties, especially in the stratosphere. It is therefore necessary to evaluate the consistency among SWV distributions simulated using different input reanalysis products. In this study, we evaluate the representation of SWV and its variations on multiple timescales using simulations over the period 1980–2013. Our simulations are based on the Chemical Lagrangian Model of the Stratosphere (CLaMS) driven by horizontal winds and diabatic heating rates from three recent reanalyses: ERA-Interim, JRA-55 and MERRA-2. We present an inter-comparison among these model results and observationally-based estimates, using a multiple linear regression method to study the annual cycle (AC), the quasi-biennial oscillation (QBO), and longer-term variability in monthly zonal-mean $H_2O$ mixing ratios forced by variations in the El Niño–Southern Oscillation and the volcanic aerosol burden. We find reasonable consistency among simulations of the distribution and variability of SWV with respect to the AC and QBO. However, the amplitudes of both signals are systematically weaker in the lower and middle stratosphere when CLaMS is driven by MERRA-2 than when it is driven by ERA-Interim or JRA-55. This difference is primarily attributable to relatively slow tropical upwelling in the lower stratosphere in simulations based on MERRA-2. Two possible contributors of the slow tropical upwelling in the lower stratosphere are found to be the large long-wave radiative effect and the unique assimilation process in MERRA-2. The impacts of ENSO and volcanic aerosol on $H_2O$ entry variability are qualitatively consistent among the three simulations despite differences of 50–100% in the magnitudes. Trends show larger discrepancies among the three simulations. CLaMS driven by ERA-Interim produces a neutral to slightly positive trend in $H_2O$ entry values over 1980–2013 ($+0.01$ ppmv decade$^{-1}$), while both CLaMS driven by JRA-55 and CLaMS driven by MERRA-2 produce negative trends but with significantly different magnitudes ($-0.22$ ppmv decade$^{-1}$ and $-0.08$ ppmv decade$^{-1}$, respectively).





# 1 Introduction

Water vapor is one of the most influential greenhouse gases (Forster and Shine, 1999), modulating not only the surface radiative forcing (Forster and Shine, 2002; Solomon et al., 2010; Riese et al., 2012) but also stratospheric ozone loss (e.g. Vogel et al., 2011). The extreme dryness of the stratosphere results from 'freeze-drying' of air entering the stratosphere, as initially explained in relation to the Brewer-Dobson circulation (BDC; Brewer, 1949). Mixing ratios of water vapor in the lower stratosphere are extremely low as a result, but nonetheless vary substantially in time and space.

There remain substantial inconsistencies among observational estimates of stratospheric water vapor (SWV), including both balloon-based and satellite-based instruments (e.g. Hegglin et al., 2014; Lossow et al., 2017b; Khosrawi et al., 2018), which are even more pronounced among atmospheric reanalyses (Davis et al., 2017). Uncertainties in balloon- or aircraft-based observations arise primarily due to insufficient spatiotemporal coverage (Müller et al., 2016) and the unreliability of operational sondes at stratospheric altitudes (e.g. Dirksen et al., 2014). Inconsistencies in satellite observations of SWV reflect limited observational periods and short overlap times (Hegglin et al., 2014), which make it difficult to control for platform-specific biases or differences in temporal or spatial sampling patterns (Lossow et al., 2017b; Khosrawi et al., 2018). Observations of SWV are rarely assimilated in reanalysis systems, for which estimates of SWV are effectively model products, in some cases nudged to climatologies (Fujiwara et al., 2017). Differences in $H_2O$ observations in the upper troposphere and stratosphere and the unreliability of reanalysis estimates of SWV have motivated several efforts to merge observational records from different satellites (Hegglin et al., 2014; Davis et al., 2016). Such homogenized records facilitate analysis of long-term changes in SWV across the most recent 2–3 decades.

Chemical transport models (CTMs) provide another approach to understanding the multi-timescale variability and global distribution of SWV (e.g. Schoeberl et al., 2012; Tao et al., 2015). Water vapor values entering the stratosphere are determined primarily by the lowest temperature along their advective transport pathway in the upper troposphere (Mote et al., 1996; Gettelman et al., 2000). Lagrangian approaches provide more accurate records of the temperature histories of air parcels compared to Eulerian models, and therefore provide more reliable representations of entry mixing ratios in SWV (e.g. Fueglistaler et al., 2005; Schoeberl et al., 2012). This distinction is particularly relevant around the tropopause, where temperature gradients are both large and highly variable (Fueglistaler et al., 2009a). In addition to transport across the tropical tropopause, photochemical oxidation of methane ($CH_4$) is an important source of SWV, especially in the middle and upper stratosphere. Previous studies have concluded that recent increases in $CH_4$ have substantially contributed to long-term variability of $H_2O$ in the stratosphere (e.g. Rohs et al., 2006; Hurst et al., 2011; Hegglin et al., 2014). In this study, we use a forward Lagrangian transport model with implanted methane chemistry to study the climatological features of SWV, and compare the results against observational estimates from the Stratospheric Water and OzOne Satellite Homogenized (SWOOSH) dataset (Davis et al., 2016) and the Aura Microwave Limb Sounder (MLS) (Livesey et al., 2017).

Key aspects affecting the performance of CTMs with respect to SWV are the meteorological fields selected to drive the transport and freeze-drying in the tropical tropopause layer (TTL). Modern meteorological reanalyses, as 'best estimates' of the historical evolution of the atmospheric state (Fujiwara et al., 2017), are commonly used to provide the necessary meteoro-



logical variables for CTMs. Current widely-used reanalysis products include the European Centre for Medium-range Weather Forecasting (ECMWF) Interim Reanalysis (ERA-I; Dee et al., 2011), the Japanese 55-yr Reanalysis (JRA-55; Kobayashi et al., 2015), the Modern-Era Retrospective Analysis for Research and Applications Version 2 (MERRA-2; Gelaro et al., 2017), and the Climate Forecast System Reanalysis (CFSR; Saha et al., 2010). Substantial uncertainties among these reanalyses have

been identified by previous studies, including significant differences in temperature and wind structures (e.g. Mitchell et al., 2015; Kawatani et al., 2016; Long et al., 2017), diabatic heating rates in the tropical upper troposphere and lower stratosphere (UTLS) (e.g. Fueglistaler et al., 2009b; Wright and Fueglistaler, 2013), and representations of the BDC (e.g. Abalos et al., 2015). Such differences among reanalysis products are critical for simulations of atmospheric composition but have rarely been discussed in this context.

Schoeberl et al. (2012) evaluated SWV simulated using the same trajectory model with meteorological fields taken from MERRA (the predecessor of MERRA-2; Rienecker et al., 2011), ERA-I, and CFSR. The trajectory model based on ERA-I produced a low bias of $H_2O$ in the lower stratosphere and a 'tape-recorder' signal (Mote et al., 1996) that was 30% too fast. By contrast, use of MERRA resulted in reasonable $H_2O$ entry values but a tape-recorder signal that was 30% too slow, while use of CFSR resulted in a wet bias in $H_2O$ entry values but a reasonable propagation of the tape-recorder signal. An earlier

study by Wright et al. (2011) reported similar biases in simulated water vapor vapor entering the stratosphere via the Asian monsoon, with trajectories based on MERRA indicating systematically larger entry mixing ratios relative to trajectories based on ERA-I. Further evaluation of how uncertainties in current reanalysis products impact simulations of $H_2O$ in the stratosphere is lacking, particularly with respect to the more recently released JRA-55 and MERRA-2 products. The former is currently the most recent full-input reanalysis ( which assimilate the full observing system) to provide coverage of the pre-satellite era, while

the model used for the latter has been reported to have a more realistic spontaneous QBO than its predecessor MERRA (Coy et al., 2016).

In this study, we provide an intercomparison of SWV produced using a Lagrangian transport model driven by three recent reanalysis products: ERA-I, MERRA-2 and JRA-55. We also present comparisons against SWV estimates from satellite observations. Our evaluations of simulated SWV focus on the climatological annual mean, the annual cycle (AC) and the

quasi-biennial oscillation (QBO). We also discuss some key sources of variability, including the El Niño–Southern Oscillation (ENSO) and variations in volcanic aerosol, as well as long-term linear trends in $H_2O$ entry mixing ratios. The objective of this work is to evaluate the sensitivity of simulated $H_2O$ variability to uncertainties in the driving meteorological fields among recent reanalysis products. The study also sheds light on the quality of reanalysis products in the stratosphere, especially their representations of dynamical fields (e.g. winds, heating rates and temperatures). Few independent observational datasets

are available to evaluate these dynamical variables in the upper troposphere and stratosphere, especially with respect to their influences on SWV.





## 2 Data and Methods

### 2.1 CLaMS simulations

Our simulations of SWV cover the period 1980–2013 using the Chemical Lagrangian Model of the Stratosphere (CLaMS;
McKenna et al., 2002; Konopka et al., 2004; Pommrich et al., 2014). Modelled water vapor mixing ratios entering the strato-
sphere are based on a simplified dehydration scheme as described in details by Poshyvailo et al. (2018). If saturation occurs
along a CLaMS air parcel trajectory, the amount of water vapor in excess of the critical saturation mixing ratio is instanta-
neously transformed to the ice phase. Ice sedimentation is then parameterized as a function of mean ice particle radius and the
corresponding fall speed (Hobe et al., 2011; Ploeger et al., 2013). Furthermore, if ice exists in a sub-saturated parcel, this ice
is instantaneously evaporated until saturation is reached. Modelled water vapor mixing ratios are influenced by parameterized
small-scale mixing driven by large-scale deformation (McKenna et al., 2002; Konopka et al., 2004). Poshyvailo et al. (2018)
tested the sensitivity of modelled $H_2O$ entry mixing ratio to mixing parameters in CLaMS and found that a greater intensity
of small-scale mixing leads to moistening of the stratosphere ($\sim$0.5ppmv) and amplification of the annual cycle ($\sim$0.2ppmv,
approximately half of peak-to-peak annual cycle). Our study uses the reference configuration of mixing parameters, which has
been shown to produce the best agreement with MLS observations (e.g. Tao et al., 2015; Konopka et al., 2016; Poshyvailo
et al., 2018).

Temperatures, horizontal winds and diabatic heating rates are prescribed from the ERA-I (Dee et al., 2011), JRA-55
(Kobayashi et al., 2015), and MERRA-2 (Gelaro et al., 2017) reanalyses. Water vapor mixing ratios at pressures greater
than $\sim$500 hPa are set equal to water vapor mixing ratio products from the corresponding reanalysis. Additional information
regarding the structures of these reanalysis systems has been provided by Fujiwara et al. (2017). We exclude CFSR because
the model and data assimilation system used to produce this reanalysis changed abruptly at the beginning of 2011 (Fujiwara
et al., 2017). This and other stream transitions produced substantial discontinuities in stratospheric variables based on CFSR
during the 1980–2013 analysis period (Davis et al., 2017; Long et al., 2017).

We use the critical saturation rate of 100% with respect to ice, although supersaturation with respect to ice is frequently
observed at these altitudes (e.g. Krämer et al., 2009). Within CLaMS, we have the freedom to 'optimize' supersaturation
thresholds to produce better agreement with observed $H_2O$ values. The effects of tuning the critical supersaturation threshold
in CLaMS are comparable to those of applying a frost point offset to the Lagrangian dry point temperature; i.e. an increase
in the supersaturation threshold enhances both the mean value and the amplitude of the annual cycle in simulated $H_2O$ (Liu
et al., 2010; Fueglistaler et al., 2013). Although this aspect of the CLaMS configuration could be optimized for each reanalysis
temperature field by comparing CLaMS output with MLS or HALOE observations, we have not applied such optimizations
before conducting this study (for the case of ERA-I, see Ploeger et al., 2013). As a consequence, comparison of modelled values
of $H_2O$ entry mixing ratio against observations cannot unequivocally validate the quality of Lagrangian dry point temperatures
in these reanalyses (Fueglistaler et al., 2013). Our intercomparison among modelled and observed $H_2O$ values is thus limited
to uncertainties related to using different reanalyses under the standard configuration of the model.





Methane oxidation is included as a source of water vapor mainly in the middle and upper stratosphere, with concentrations of hydroxyl, atomic oxygen, and chlorine radicals taken from model-generated climatologies (Pommrich et al., 2014). We explicitly diagnose the fraction of water vapor supplied by methane oxidation. Methane-supplied water vapor ($H_2O_{CH_4}$) at any given location is calculated as

$$H_2O_{CH_4} = 2 \cdot (CH_4{}^{rec} - CH_4) = 2\alpha CH_4{}^{rec}, \tag{1}$$

where $CH_4$ is modelled methane accounting for loss by oxidation and $CH_4{}^{rec}$ is the passively-transported methane assuming the same source and circulation without photochemical loss. Reconstructed methane values ($CH_4{}^{rec}$) are calculated as the mean tropospheric $CH_4$ prescribed at the lower boundary of the model (see Figure B1 in Appendix B), lagged by the mean age of air ($\Gamma$) in each run (Ploeger et al., 2015):

$$CH_4{}^{rec}(x,t) = CH_4{}^{LB}(t - \Gamma(x,t)). \tag{2}$$

The coefficient $\alpha$ in eq. (1) is the fractional release factor of $CH_4$. Ostermöller et al. (2017) showed that chemical loss can increase the time dependence of $\alpha$. Our calculation of $\alpha$ neglects this effect, as Ostermöller et al. (2017) also suggested that this influence should be very small for tracers with weak tropospheric trends, like $CH_4$ ($\sim$0.2-0.3% $yr^{-1}$).

Stratospheric water vapor without the contribution from methane oxidation ($H_2O_{nCH_4}$) is then diagnosed as

$$H_2O_{nCH_4} = H_2O - H_2O_{CH_4}. \tag{3}$$

Our intercomparison of the three reanalysis-driven CLaMS runs covers the period from January 1980 through December 2013 (hereafter referred to as the 'CLaMS period') within the vertical range from $\theta$=350 K to $\theta$=2000 K. A reference monthly-mean SWV is calculated by averaging the three reanalysis-driven runs (hereafter referred to as the Multi-Reanalysis Mean or MRM). The MRM is used as a benchmark to more effectively highlight differences among the three model runs.

## 2.2 Observational estimates

The Stratospheric Water and Ozone Satellite Homogenized (SWOOSH) database provided by the NOAA Chemical Sciences Division (CSD) contains vertically resolved ozone and water vapor data from the SAGE-II/III, UARS HALOE, UARS MLS, and Aura MLS satellite instruments starting from 1984 (Davis et al., 2016). We use the SWOOSH zonal-mean monthly-mean time series of merged water vapor mixing ratios with 2.5° resolution on 21 isentropic levels from 300 K to 650 K. The homogenization process, which has been described by Davis et al. (2016), is designed to minimize artificial jumps in time and account for inter-satellite biases. The merged SWOOSH data thus provide a long-term time series with reliable representations of interannual to decadal variability.

We compare monthly mean water vapor from CLaMS runs with SWOOSH water vapor on the same vertical grid (21 $\theta$ levels from 350 K to 650 K). For each latitude–$\theta$ grid location, comparison between CLaMS and SWOOSH starts from the first month when SWOOSH has more than 12 months of available $H_2O$ data within the following 2 years and ends in December 2013. In the following, we refer to this period as the 'SWOOSH period'. Note that while CLaMS provides continuous temporal





coverage, the SWOOSH data may include some gaps. Additional details regarding SWOOSH data coverage are provided in Appendix A.

In addition to SWOOSH, we use Aura MLS version 4 retrievals of $H_2O$ (Livesey et al., 2017) for comparison with CLaMS simulations during the period 2005–2013 (the 'MLS period'). MLS provides over 3000 profiles per day, with water vapor

estimates at 30 pressure levels from 316 hPa to 1 hPa. The vertical resolution in the stratosphere is approximately 3 km (2.5–3.5 km). Uncertainties in the water vapor retrievals are of the order of ∼10% in the lower stratosphere and ∼5% in the upper stratosphere. The relatively high frequency of horizontal sampling and high quality of vertical profiles allows MLS $H_2O$ to reliably represent the zonally- and monthly-averaged distribution of SWV. We interpolate MLS $H_2O$ profiles to 26 isentropic levels, chosen to span the range 350 K–2000 K at a vertical resolution close to that of the original retrievals.

The SWOOSH dataset is also based in part on Aura MLS version 4 retrievals, particularly during the MLS period. Thus, the comparisons of CLaMS against SWOOSH and comparisons of CLaMS against MLS during the MLS period are not independent. Differences between the comparisons are due to the homogenization procedure applied in SWOOSH. We did not apply the MLS averaging kernels to CLaMS $H_2O$ simulations. It is because the application of MLS averaging kernels to CLaMS $H_2O$ simulations can potentially produce artefacts, especially at high latitudes (more details provided by Ploeger et al.,

2013, their Fig. 2). Our main focus is on the comparison among the reanalyses and to apply averaging kernels could smear out the differences.

### 2.3    Extraction of variability at multiple timescales

The objective of this study is to examine and compare the climatological features of variability in model-based and observationally-based estimates of SWV. Pronounced periodic signals in SWV include the annual cycle (AC), the Quasi-Biennial Oscillation

(QBO) and the Semi-Annual Oscillation (SAO). The AC, QBO and SAO are therefore considered explicitly in our regression model:

$$\chi(t) = \bar{\chi} + \chi_{AC}(t) + \chi_{SAO}(t) + \chi_{QBO}(t) + \chi_{res}(t). \tag{4}$$

After determining the mean value calculated over the whole considered time series ($\bar{\chi}$), we extract AC ($\chi_{AC}(t)$), SAO ($\chi_{SAO}(t)$) and QBO ($\chi_{QBO}(t)$) signals following the harmonic regression method used by Lossow et al. (2017b). In this method, one sine/cosine pair terms are used for each regressions (AC and SAO), respectively. Two quasi-orthogonal QBO time

series (namely, the zonal wind signals at 50 hPa and 30 hPa over Singapore provided by Freie Universität Berlin) are used for the QBO regression. The AC, QBO and SAO variables are assessed from two aspects of periodic variations: amplitude ($A_{AC}$, $A_{QBO}$, $A_{SAO}$) and phase ($P_{AC}$, $P_{QBO}$, $P_{SAO}$). Note that 1) the amplitude represents half the corresponding variation from maximum to minimum, 2) $P_{AC}$ and $P_{SAO}$ are defined as the month of the annual maximum in the regression fit, and 3) $P_{QBO}$ is defined as the month (within 0–28) with the largest lag-correlations between the QBO fit and the 50 hPa Singapore wind.

The linear trend ($C_{trd}$) is estimated by applying a least-squares linear fit to the residuum of $H_2O$ variability ($\chi_{res}(t)$ in eq. (4)) after removing the AC, QBO and SAO signals. Note that the linear trend for $\chi_{res}(t)$ is identical to the trend of the original $H_2O$ time series since periodic signals such as AC and QBO have zero long-term trend. Additional variability after





subtracting the AC, QBO and SAO signals may result from the influences of ENSO (e.g. Randel et al., 2004; Fueglistaler and Haynes, 2005; Konopka et al., 2016; Yan et al., 2018), variations in stratospheric aerosol (e.g. Joshi and Shine, 2003; Diallo et al., 2017). The contributions of such variations to $H_2O$ entry mixing ratios are discussed in sec. 6.

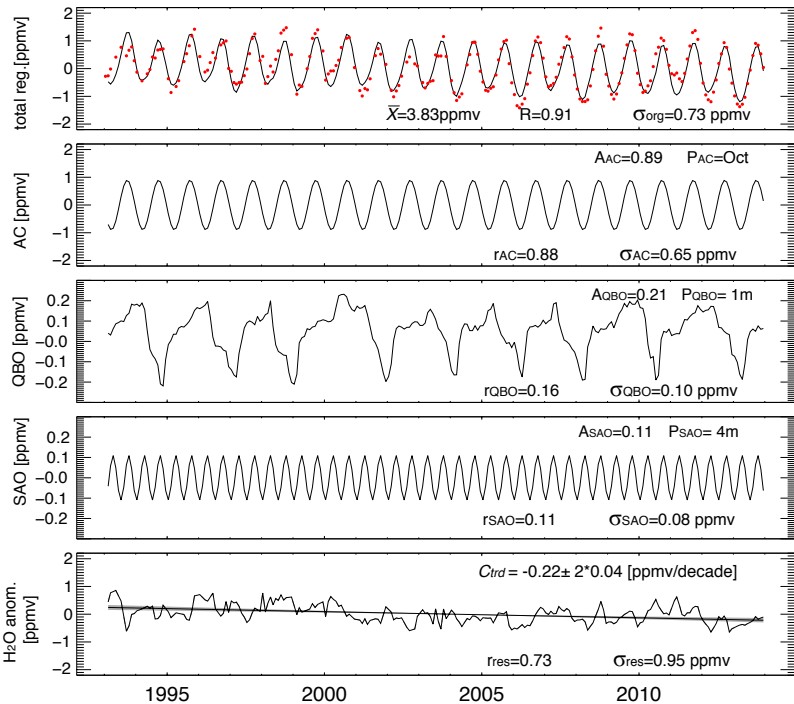

**Figure 1.** Multiple linear regression results for monthly-averaged tropical mean (20°S–20°N) water vapor on the $\theta = 400\,K$ isentropic surface from SWOOSH. The top panel shows anomalies of SWOOSH data relative to the mean value over the 1993–2013 period (red dots) and the regression fit (black line) based on eq. (4) (correlation coefficient $r = 0.91$). The AC, QBO and SAO components of the regression are shown in the following three panels, respectively. Additional variability beyond the three periodic components (residuum of the regression) is shown in the bottom panel. The linear trend in this time series is also shown, with $\pm 2\sigma$ uncertainty shown as grey shading. Note the different $y$-axis ranges across the different panels. The amplitudes and phases of the AC, QBO and SAO signals are listed in the upper right corners of the corresponding panels. The standard deviations corresponding to each contribution are listed near the bottom of each panel, together with correlation coefficients for each term against the original time series. The standard deviation of the original SWOOSH time series ($\sigma_{org}$) is 0.73 ppmv.

Figure 1 provides example regression results for SWV on the 400 K isentropic surface in the tropics (20°S-20°N) based on the SWOOSH monthly merged satellite dataset. This time series approximates that of $H_2O$ values entering the stratosphere ($H_2O_e$) at the base of the 'tropical pipe' (Plumb, 1996), and is characterized by a climatological annual mean of 3.83 ppmv and a negative trend of 0.22 ppmv decade$^{-1}$. The regression model explains over 90% of the variation in $H_2O_e$. According to the correlation coefficient of each variable, the periodic terms listed in descending order of influence are AC, QBO and then SAO.



The AC represents the most pronounced variation in $H_2O$ at this level, with a correlation coefficient of 0.88, an amplitude of $A_{AC} = 0.89$ ppmv and an annual maximum in $P_{AC} = $ October. The QBO is also a significant factor, with an amplitude of 0.21 ppmv and a maximum lag-correlation (correlation coefficient=0.16) against the 50 hPa Singapore winds of $P_{QBO} = $ 1 month. The bottom panel of Figure 1 shows other variability in $H_2O$ entry mixing ratios (the residuum of the regression in eq. (4)). This residual variability represents a substantial component of the total variability, and contains intraseasonal variability, inter-annual variability and the long-term trend. Features of the AC, QBO and the other variability are compared and discussed in more detail in section 4, section 5 and section 6, respectively. Since the contribution of SAO to the total variance is generally smaller than that of other variables, we defer discussion of the SAO influence to Apprendix C.

## 3  Climatological annual mean of stratospheric water vapor

Figure 2 shows climatological annual-mean simulated SWV from the three CLaMS runs, as well as the components of SWV with and without $CH_4$ oxidation ($H_2O_{CH_4}$ and $H_2O_{nCH_4}$). The climatological mean is based on the period 1985–2013 because $CH_4{}^{rec}$ cannot be diagnosed during the first few years of the simulation. Zonal-mean $H_2O$ matches zonal-mean $H_2O_{nCH_4}$ well in the lower stratosphere but consists mainly of $H_2O_{CH_4}$ in the upper stratosphere.

The driest stratosphere is simulated by CLaMS-ERA, while the wettest is simulated by CLaMS-JRA. This difference is mainly attributable to $H_2O_{nCH_4}$, as shown in Figure 2(A2-C2), which is in turn controlled primarily by entry mixing ratios at the tropical tropopause (with the notable exception of dehydration in the Southern Hemisphere polar vortex). These differences reflect differences in tropical tropopause temperatures among the reanalyses (black contours in middle panel; see also Figure 4). Tropical tropopause temperatures differ by about 1 K between ERA-I and JRA-55, consistent with a difference of ∼0.6 ppmv in $H_2O$ mixing ratios assuming the same 100% saturation threshold applied in the CLaMS model.

The amount of $H_2O$ from $CH_4$ oxidation is related to the time an air parcel has spent in the stratosphere. This time is measured by the stratospheric age-of-air as shown in the upper panels of Figure 3. Values of $H_2O_{CH_4}$ from the CLaMS-MRA simulation are systematically larger than those from the CLaMS-ERA or CLaMS-JRA simulations (see Fig. 2 A3-C3). The most pronounced high biases in $H_2O_{CH_4}$ and $\alpha$ in CLaMS-MRA relative to the other two simulations are co-located with the sharpest gradients in $H_2O_{CH_4}$ and $\alpha$. Figure 3 provides useful context for interpreting these differences, as high values of $H_2O_{CH_4}$ are accompanied by systematically older stratospheric air in CLaMS-MRA. Note that a consistent climatological mean age differences within the same model framework are presented in Ploeger et al. (2019). The lower panels of Fig. 3 show differences in reanalysis heating rates, which are used to drive vertical transport in the CLaMS model. Tropical upwelling in the shallow branch of the BDC is weaker in MERRA-2 than in ERA-I or JRA-55, thus producing larger gradients of AoA and $H_2O_{CH_4}$ between the lower and middle stratosphere. Quantitatively, relative differences in time-mean zonal-mean diabatic heating rates at 400–500 K are as large as 50% between MERRA-2 and ERA-I, resulting in large differences in the vertical advection of SWV and other tracers. It is worth noting that differences among the reanalyses in the lower stratosphere are larger in the NH subtropics than in the deep tropics or the SH subtropics (not shown).




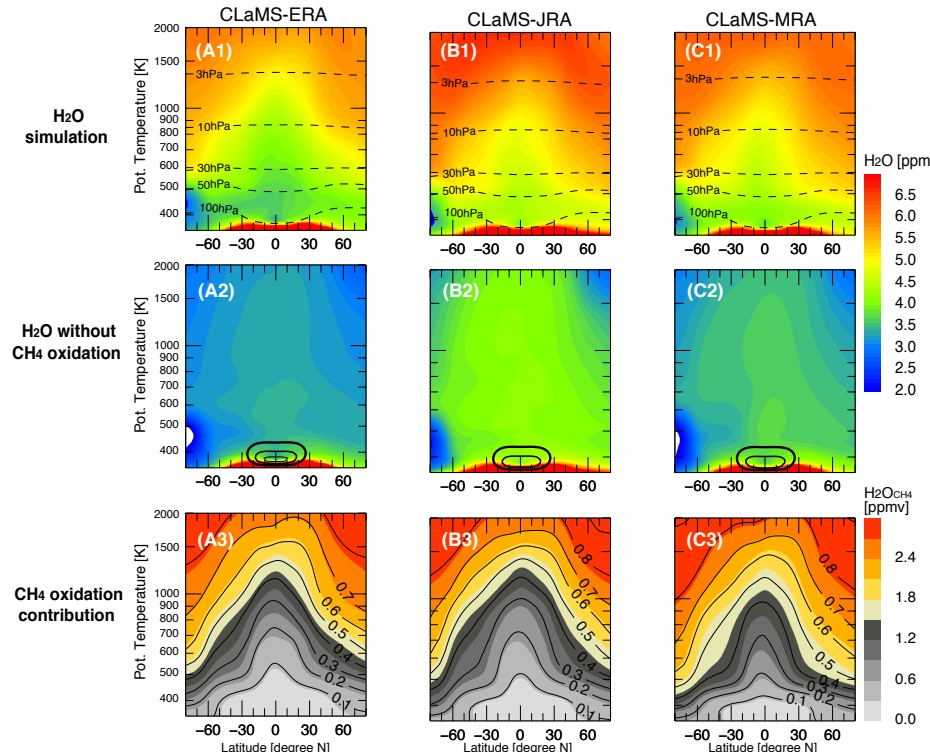

**Figure 2.** Comparison of climatological annual mean SWV during the CLaMS period (1985–2013). Top row (A1-C1): annual-mean zonal-mean simulated $H_2O$ based on (left-to-right) CLaMS-ERA, CLaMS-JRA and CLaMS-MRA. Dashed lines mark the zonal-mean locations of isobaric surfaces. Middle row (A2-C2): as in (A1-C1), but for simulated $H_2O$ without methane oxidation. These distributions highlight the effects of variations in $H_2O$ entry values (tropopause temperature). Black contours show zonal-mean temperatures of 192.5 K, 195 K and 200 K (from thin to thick lines) near the tropical tropopause. Bottom row (A3-C3): as in (A1-C1), but for simulated $H_2O$ from $CH_4$ oxidation. The distributions reflect the effect of the BDC on water vapour produced by methane oxidation. Black contours show the fractional $CH_4$ oxidation ratio ($\alpha$).





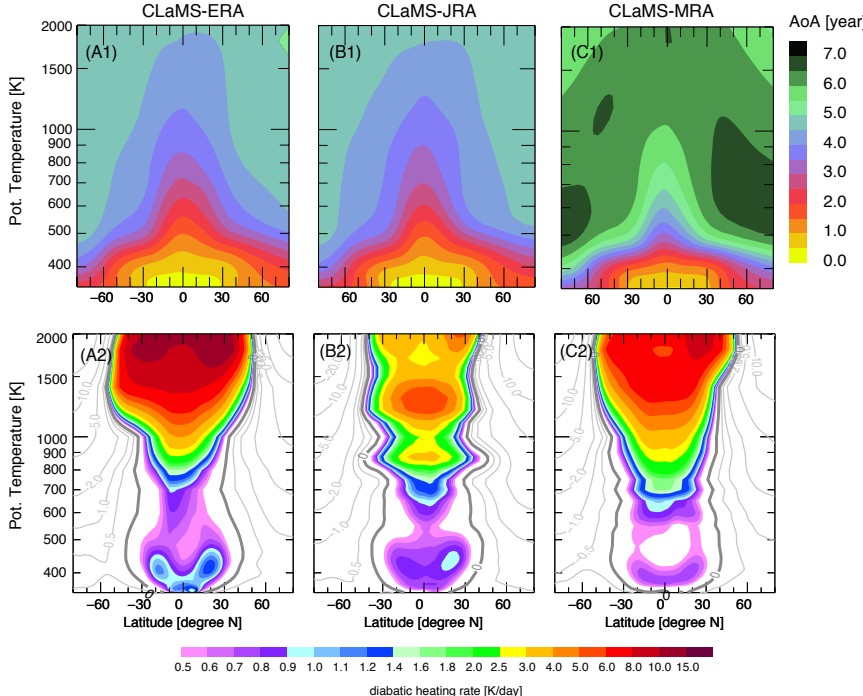

**Figure 3.** Comparison of climatological annual-mean zonal-mean age-of-air from the CLaMS-ERA, CLaMS-JRA and CLaMS-MRA simulations (A1-C1), along with corresponding cross-isentropic velocities ($\dot{\theta}$) based on total diabatic heating rates for each reanalysis product (A2-C2). Color shading in the lower panels indicates zonal-mean diabatic upwelling rates that exceed $0.5\,\mathrm{K\,day^{-1}}$. The thick grey line shows the zero-contour of diabatic heating rates. Lighter thin grey lines show diabatic downwelling.

Although diabatic upwelling in the tropical lower stratosphere is weakest in MERRA-2, upwelling in the tropical middle stratosphere ($\theta = 600\,\mathrm{K}$–$1000\,\mathrm{K}$) is stronger and covers a wider range of latitudes in MERRA-2 than in ERA-I or JRA-55. Above $\theta = 1000\,\mathrm{K}$, the deep BDC as represented in JRA-55 is noticeably weaker than that in MERRA-2 or ERA-I. Previous studies have pointed out that the mean magnitudes of tropical upwelling in the lower stratosphere are substantially different among different reanalysis products (Wright and Fueglistaler, 2013; Abalos et al., 2015).

## 4 Annual cycle

The annual cycle is the largest contributor to total variability in SWV. Moreover, seasonal variations in SWV give essential insight into the overall behavior of $H_2O$, as these variations reflect the effects of seasonal changes in both the stratospheric circulation and temperatures at the tropical tropopause.





## 4.1 The H$_2$O tape-recorder

Before discussing the global features of the annual cycle in SWV, we first report the simulated climatological seasonalities of H$_2$O entry mixing ratios (H$_2$O$_e$) from the three simulations. Values of H$_2$O$_e$ are mainly controlled by 'freeze-drying' around the tropical tropopause and the upward propagation of signals imprinted by variations in the conditions under which

this 'freeze-drying' takes place. The upward propagation of these signals produces the well-known stratospheric tape-recorder structure described by Mote et al. (1996). In this study, we define water vapor entry mixing ratios (H$_2$O$_e$) as averages over the 20°S–20°N tropical band on the $\theta = 400$ K isentropic surface.

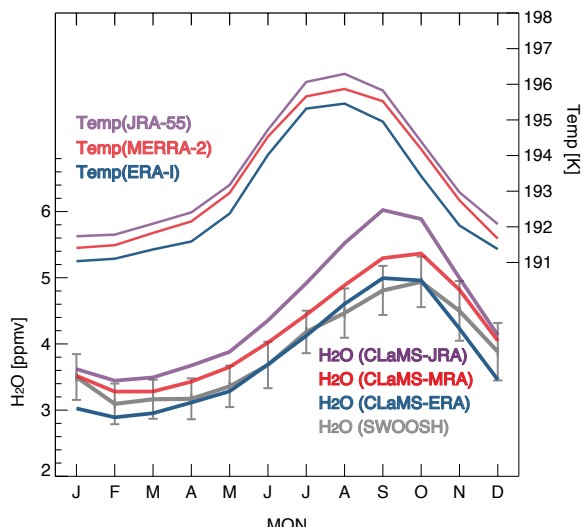

**Figure 4.** Relationships between tropical tropopause temperatures (minimum temperature between $\theta = 360$ K and $420$ K averaged over 20°S-20°N) and water vapor entry mixing ratios (mean H$_2$O mixing ratios on $\theta = 400$ K averaged over 20°S-20°N). The upper set of lines shows the climatological annual cycles for tropical tropopause temperatures, while the lower set of lines shows the climatological annual cycles for water vapor entry mixing ratios.

    Figure 4 summarizes the relationship between the climatological annual cycles of tropical tropopause temperatures (upper set of lines) and H$_2$O$_e$ (lower set of lines). The annual cycles in tropical mean tropopause temperatures based on the three

reanalyses (upper set of lines) are similar in both amplitude and phase, but with $\pm 1$ K differences in mean value. Mean H$_2$O$_e$ differences among the three CLaMS runs are roughly consistent with the corresponding differences in mean tropopause temperatures, with a 1 K difference in temperature corresponding to a $\sim$0.6 ppmv difference in H$_2$O entry mixing ratios. Although the differences in the mean tropical tropopause temperature are consistent with the H$_2$O differences, it is actually the Lagrangian cold point controls the H$_2$O entry values and makes the H$_2$O differences, which will be indicated by the next figure. Compared

with SWOOSH, simulated annual cycles of H$_2$O$_e$ from CLaMS-ERA and CLaMS-MRA are relatively reasonable in terms of



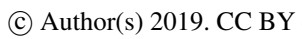


both absolute value and amplitude (i.e. generally within uncertainties). By contrast, $H_2O_e$ from CLaMS-JRA is systematically larger than that indicated by SWOOSH or the other two runs. Simulated values based on CLaMS-JRA are $\sim$0.5 ppmv larger than SWOOSH estimates in February and $\sim$1 ppmv higher than SWOOSH estimates in September. Thus, CLaMS-JRA produces $\sim$0.5 ppmv larger peak-to-peak annual amplitude than other estimates. This is qualitatively consistent with non-linearity

in the Clausius-Clapeyron equation, which mean that the effects of temperature offsets on $H_2O$ values are larger when the average temperature is higher (Liu et al., 2010; Fueglistaler et al., 2013). Thus, a larger annual amplitude in entry mixing ratios is expected when the average tropopause temperature is larger. Moreover, this makes it possible that an optimization of the dehydration scheme could effectively adjust not only the modelled mean entry mixing ratio but also the amplitude of the annual cycle to better match observations.

The annual cycle in tropical tropopause temperature leads that of $H_2O_e$ by 1–2 months because ascent within the tropical tropopause layer is relatively slow. Given the strong consistency among tropopause temperature annual cycle phases as represented in the reanalyses, we find some notable differences in the phases of entry $H_2O$ annual cycle. CLaMS-ERA and CLaMS-JRA indicate very consistent phases of the AC, with the annual minimum occurring in February and the annual maximum occurring in September. However, the AC phase of $H_2O$ entry values based on CLaMS-MRA is in better agreement with

the SWOOSH data. The maximum value of $H_2O_e$ based on both of these estimates occurs in October, one month later than the maximum values based on CLaMS-ERA and CLaMS-JRA. This phase shift is related to the fact that the ascent in the tropical tropopause layer is slower in MERRA-2 compared to JRA-55 or ERA-I (see also comparison of MERRA and ERA-I diabatic heating rates by Wright and Fueglistaler, 2013).

In addition to mean tropical tropopause temperature, the spatial distribution of climatological tropopause temperatures is

also an important factor in determining the mean entry mixing ratio. Figure 5 shows this spatial pattern based on MERRA-2 during boreal winter (DJF) and boreal summer (JJA), along with differences in tropopause temperatures from ERA-I and JRA-55 relative to those in MERRA-2. Differences in tropopause temperatures among the three reanalysis products vary considerably with both location and season. ERA-I shows relatively large negative differences ($\leq -0.5$ K) relative to MERRA-2 over the tropical Indian and Pacific Oceans during DJF. Differences are particularly notable over the tropical western Pacific,

where tropopause temperatures based on ERA-I are substantially colder than those based on MERRA-2 during boreal winter. Tropopause temperatures in this region are widely believed to play a key role in determining the value of $H_2O_e$ during boreal winter (Randel et al., 2004; Fueglistaler et al., 2005). Warmer tropopause temperatures in JRA-55 during JJA are located mainly in the Southern Hemisphere subtropics, as well as over the Bay of Bengal and tropical Pacific, with differences relative to MERRA-2 reaching magnitudes of $\sim$1K in these regions. The latter two regions are also widely recognized as playing

influential roles in troposphere-to-stratosphere transport and the final dehydration of air parcels entering the stratosphere during boreal summer (Gettelman et al., 2004; Bannister et al., 2004; Wright et al., 2011).

A complementary picture of $H_2O$ entry values is provided in Fig. 6, which depicts the tape-recorder signal in SWV averaged over 20°S-20°N. Upward propagation of the tape-recorder signal between 450 K and 600 K is 0.5–1.5 months faster in CLaMS-ERA and CLaMS-JRA relative to SWOOSH, and 1–1.5 months slower in CLAMS-MRA than in SWOOSH. Similarly, the

amplitude of the tape-recorder signal is systematically stronger than SWOOSH in CLaMS-ERA and CLaMS-JRA, but weaker





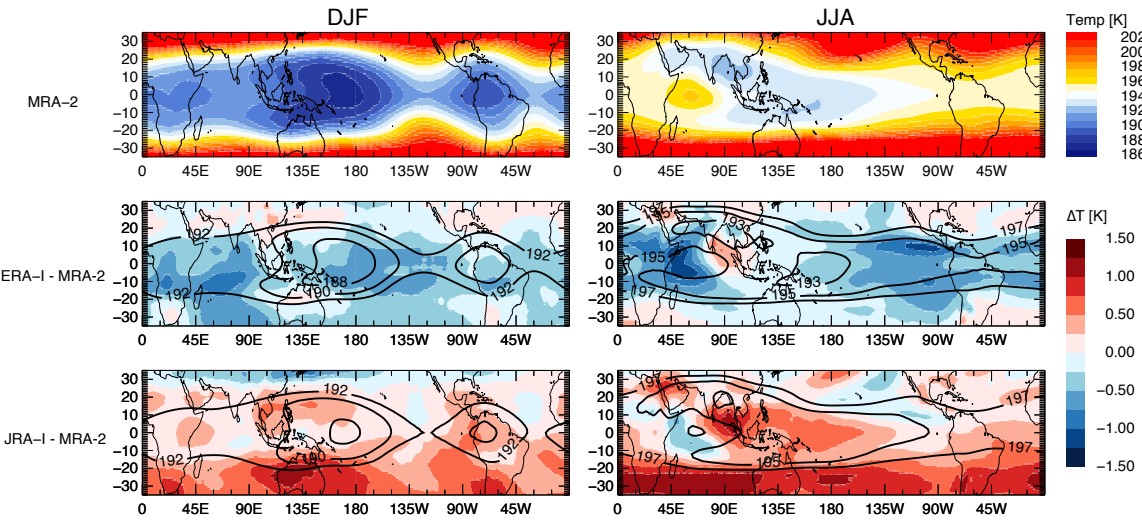

**Figure 5.** Climatological mean cold point tropopause temperatures during boreal winter (DJF, left) and boreal summer (JJA, right) from MERRA-2 (top row), along with differences in ERA-I (middle row) and JRA-55 (bottom row) relative to MERRA-2. Isolines in the middle and lower rows show absolute values of tropopause temperature. Cold point tropopause temperatures are defined here as minimum temperatures between $\theta = 360$ K and $\theta = 420$ K.

above 450 K in CLaMS-MRA. These differences are attributable in part to the relatively slow upwelling in MERRA-2 (as shown in Fig. 3). Slower upwelling not only delays the propagation of the signal but also allows more time for horizontal advection and mixing of the middle latitude air into the tropics, which tend to damp the signal. We can also see the remarkably strong contribution of $CH_4$ oxidation in CLaMS-MRA, which is shown by the blue and red contour lines in Fig. 6. The

contribution of $H_2O_{CH_4}$ to the tape-recorder signal is substantially larger in CLaMS-MRA than in the other two runs. This feature is a secondary effect of the slow tropical upwelling (in addition to more in-mixing from the extratropics), resulting in a relatively pronounced seasonal cycle in $H_2O_{CH_4}$ in CLaMS-MRA with a maximum amplitude of $\sim 0.05$ ppmv near the 450 K isentrope. The amplitude of $H_2O_{CH_4}$ in CLaMS-MRA is twice as large as that in CLaMS-JRA. Meanwhile, CLaMS-ERA shows virtually no anomalies in $H_2O_{CH_4}$ at these levels due to relatively rapid rates of ascent in the lower branch of the BDC.

Seasonal variations in $H_2O_{CH_4}$ are opposite in phase relative to seasonal variations in $H_2O_e$, and account for $\sim 20\%$ of the reduced amplitude of the $H_2O$ tape-recorder above 450 K in CLaMS-MRA.

## 4.2   Global features of the $H_2O$ annual cycle

We now consider differences in the representation of the annual cycle in $H_2O$ throughout the global stratosphere. The amplitude of the simulated annual cycle, $A_{AC}$, and its phase, $P_{AC}$, are shown in Figure 7 and Figure 8. Here, we examine the CLaMS

simulations in the context of Aura MLS observations during the MLS period (2005–2013). In general, these comparisons





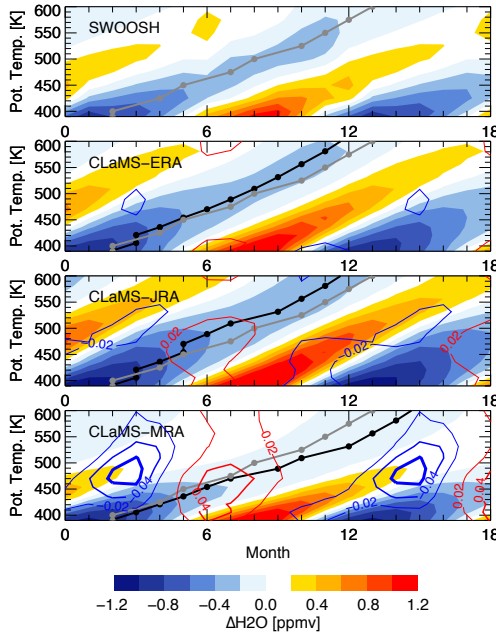

**Figure 6.** Climatological structure of the stratospheric tape-recorder signal based on SWOOSH and the three CLaMS runs. The tape-recorder is defined as anomalies in tropical ($20°$S-$20°$N) mean $H_2O$ relative to the climatological mean at each level (color shading). The phase of upward propagation (solid circles connected by a line) is defined by the largest correlation with the layer below. For ease of comparison, propagation based on SWOOSH is marked in each panel using gray circles connected by a gray line. Red and blue contours indicate positive and negative contributions of $CH_4$ to $H_2O$ anomalies (in units of ppmv, at intervals of 0.02 ppmv).

show good agreement between the CLaMS simulations and MLS. Results for comparisons against SWOOSH $H_2O$ between $\theta = 350\,K$ and $\theta = 650\,K$ are qualitatively similar to those against MLS and are not shown here.

The enhanced amplitude of the AC in 'region 1' (see Fig. 7A1 and A2) reflects the tape-recorder signal in the tropical lower stratosphere, which propagates from tropopause to the middle stratosphere and extends into subtropics and mid-latitudes.

5   Significant differences in both the phase (Fig. 8) and the amplitude of the AC are evident in the transition layer between the tropical lower stratosphere and the tropical middle stratosphere ($\theta = 450$–$500\,K$). These differences are again linked to discrepancies in the strength of tropical upwelling described above (lower panels of Fig. 3 and corresponding discussion). Consistent with our intercomparison of tape-recorder amplitudes shown in the section 4.1, the amplitude of the AC between $\theta = 450\,K$ and $\theta = 500\,K$ is larger in CLaMS-JRA and smaller in CLaMS-MRA relative to that inferred from MLS observations

10  (see dots over-plotted on Fig. 7B1-D1). Examination of the phase of the AC likewise confirms that AC-related signals propagate faster in CLaMS-ERA and CLaMS-JRA and slower in CLaMS-MRA than indicated by MLS observations (see arrows over-plotted on Fig. 8B1-D1). Although we use different methods to estimate the AC amplitude in this section relative to section 4.1, both approaches produce similar results with respect to differences among the simulated AC signals in 'region 1'.





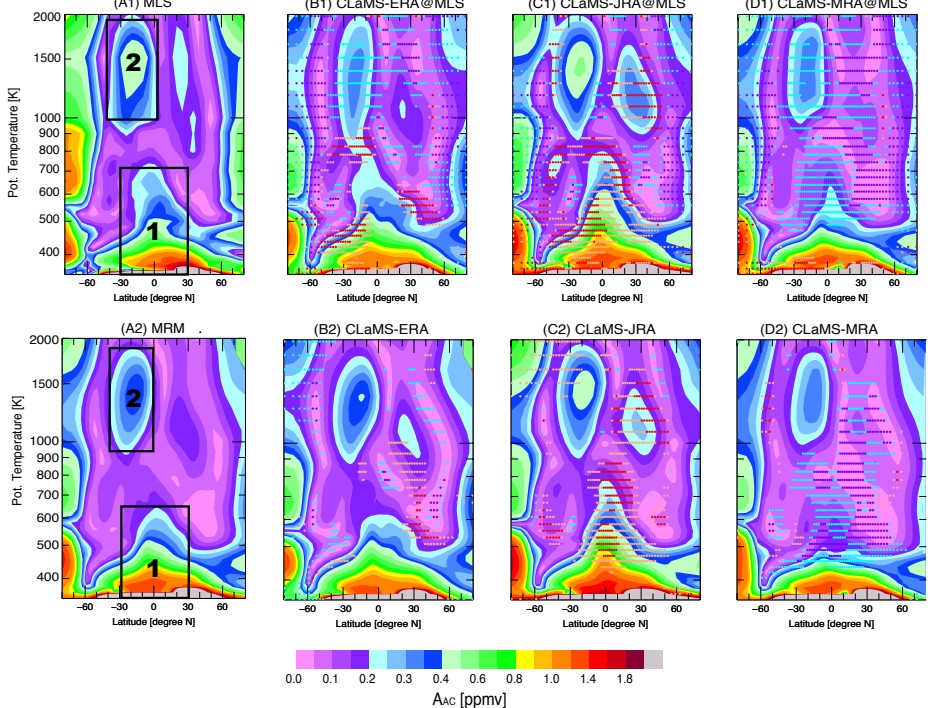

**Figure 7.** AC amplitudes during the MLS period ($\sim$2005–2013, upper row) and the entire CLaMS period (1980–2013, lower row). The leftmost panels show $A_{AC}$ based on Aura MLS observations (A1) and the mean of the three reanalysis-driven models (MRM, A2). The corresponding $A_{AC}$ values from CLaMS-ERA, CLaMS-JRA and CLaMS-MRA are shown in the rightmost three panels of each row. Dots over-plotted in panels (B1)-(D1) and (B2)-(D2) show differences in simulated $A_{AC}$ relative to the benchmark estimate shown in the corresponding leftmost panel. Light and dark red dots show positive relative differences more than 30% and 60%, respectively, while cyan and purple dots show negative relative differences less than –30% and –60%.

MLS observations further indicate a hemispheric asymmetry in the AC amplitude in 'region 1' (NH > SH), possibly related to the transport of relatively moist air from the ASM anticyclone to the stratosphere during boreal summer (e.g. Bannister et al., 2004; Wright et al., 2011). This feature is also evident in all three model-based estimates of $H_2O$. However, all three CLaMS runs overestimate the amplitude of the AC in the SH subtropics (around 380 K–450 K). One potential reason is that water vapor in the Southern Hemisphere subtropical lower stratosphere is highly sensitive to small-scale mixing processes that must be parameterized in the model (Poshyvailo et al., 2018).

Additional local maxima in the AC amplitude are found in the subtropical upper stratosphere of both hemispheres, marked as 'region 2' in Fig. 7. The AC phase in 'region 2' (Fig 8) indicates that this feature propagates quasi-meridionally. Both the enhanced amplitude of the AC and the hemispheric asymmetry (SH > NH) in this part of the stratosphere are related to seasonal variations in the deep branch of the BDC (Lossow et al., 2017a). The amplitude of the feature in 'region 2' is largest



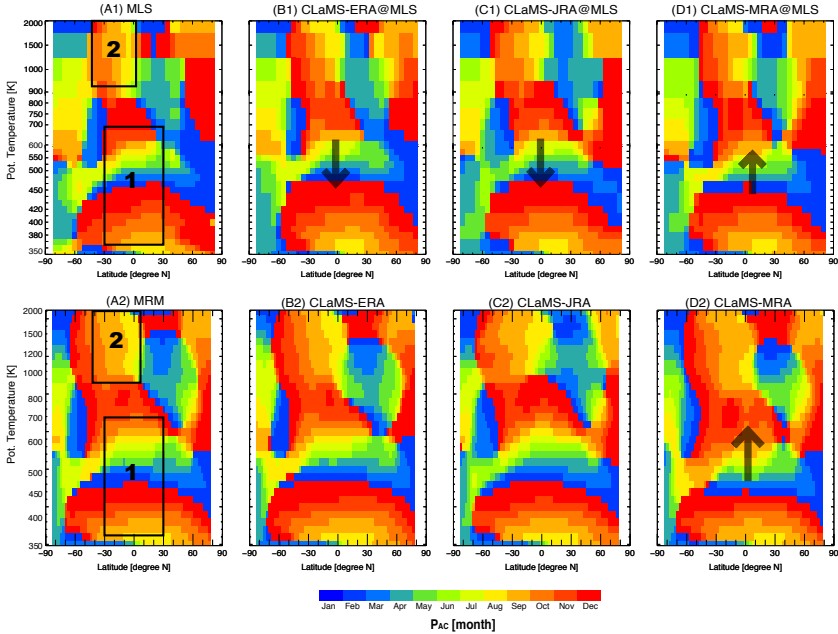

**Figure 8.** AC phases corresponding to the amplitudes shown in Fig. 7. Arrows indicate major phase differences (more than 1 month) against the benchmark estimate of $P_{AC}$ shown in the leftmost panel of the corresponding row. Upward arrows show phases that lag behind the benchmark phase while downward arrows show phases that lead the benchmark phase.

in CLaMS-JRA, followed by CLaMS-ERA and CLaMS-MRA. The amplitude implied by the MLS observations falls between those produced by CLaMS-JRA and CLaMS-ERA, and is approximately double the amplitude produced by CLaMS-MRA. Thus, the relative strength of this AC signal in the subtropical upper stratosphere is consistent with the respective strength of diabatic upwelling in the tropical lower and middle stratosphere (see Fig. 3), which is strongest in CLaMS-JRA and weakest

5   in CLaMS-MRA. Weaker upwelling in the lower part of the BDC ascending branch in CLaMS-MRA produces strong vertical gradients in tracer concentrations (including both $H_2O$ and $CH_4$) in the lower stratosphere and weak gradients in the upper stratosphere (see also Fig. 2).

## 5   The Quasi-Biennial Oscillation

We now shift our focus to another periodic oscillation in SWV: the Quasi-Biennial Oscillation (QBO). Zonal-mean distributions

10   of the climatological amplitudes and phases of the QBO are shown in Fig. 9 and Fig. 10, respectively. As above, metrics based on MLS observations are provided in the upper row and metrics based on the MRM are provided in the lower row of each figure. Three regions show clear peaks in $A_{QBO}$: the tropical lower stratosphere (region 1), the subtropical middle stratosphere (dashed square) and the tropical upper stratosphere (region 2).





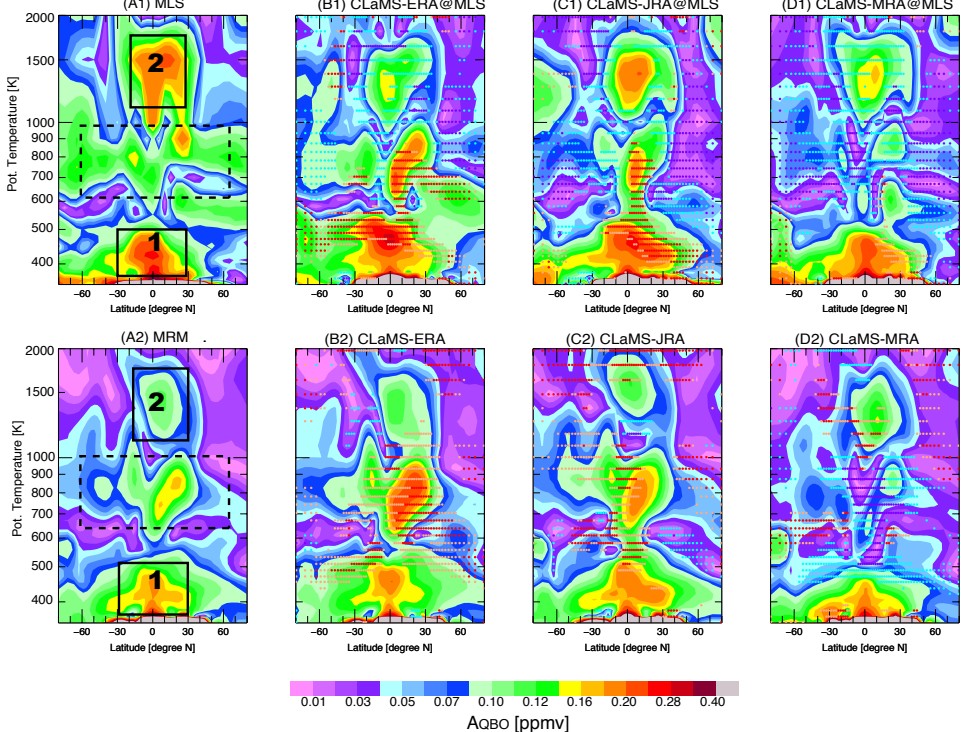

**Figure 9.** Same as Fig.7 but for QBO amplitudes. Note that the scale of color bar is different from that in Fig.7.

The large values of $A_{\mathrm{QBO}}$ in region 1 are due to QBO effects on tropical tropopause temperature, with a colder tropopause during the easterly phase of the QBO and a warmer tropopause during the westerly phase of the QBO (Baldwin et al., 2001). The phase of the QBO effect at the tropopause is therefore consistent with that of the 50 hPa QBO wind. This water vapour signal then propagates upward at a speed similar to that of the canonical tape-recorder signal. In region 1, $A_{\mathrm{QBO}}$ based on CLaMS-

5  MRA compares better with $A_{\mathrm{QBO}}$ based on MLS observations than the other two simulations. CLaMS-ERA and CLaMS-JRA both overestimate $A_{\mathrm{QBO}}$ at isentropic levels between 450 K and 550 K. This overestimation evokes similar overestimates in $A_{\mathrm{AC}}$. Both biases can be traced back to strong diabatic upwelling in the lower to middle stratosphere in ERA-I and JRA-55 (Fig. 3).

The large values of $A_{\mathrm{QBO}}$ in region 2 are mainly linked to QBO-related modulation of the stratospheric circulation. The

10  corresponding phase is effectively simultaneous with the 50 hPa QBO wind, with very little spatial shift. Satellite-derived estimates of this QBO signal indicate that amplitudes are in the range of 0.2 to 0.6 ppmv (Lossow et al., 2017a). MLS $H_2O$ is on the lower end of this range with an amplitude of $\sim$0.2 ppmv. However, CLaMS runs during the MLS period produce even smaller amplitudes of 0.1 to 0.2 ppmv, below the range implied by satellite measurements. The smallest amplitude is produced





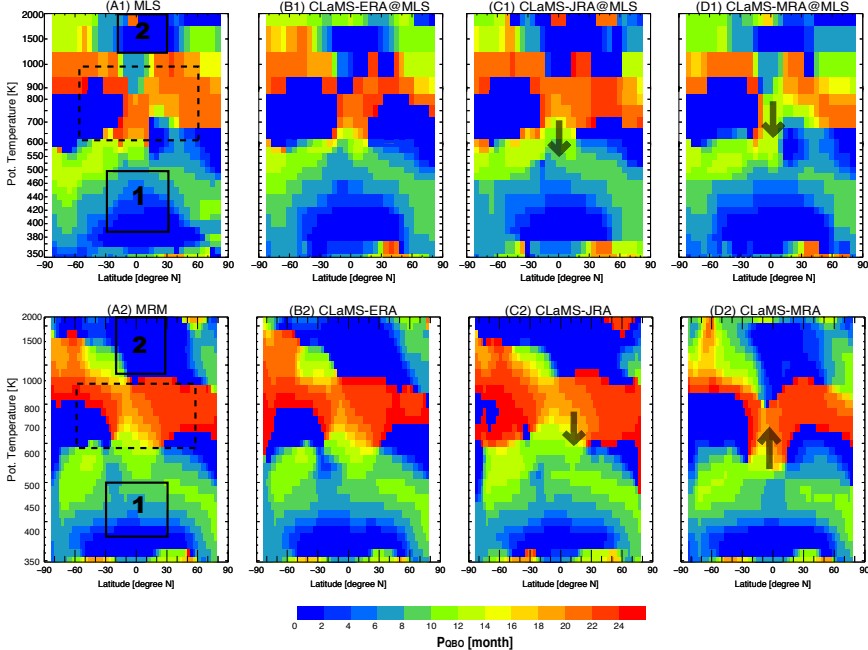

**Figure 10.** Same as Fig. 8 but for QBO phases. Arrows show major phase differences (more than 3 months) against the benchmark $P_{\mathrm{QBO}}$ shown in the leftmost panel of the corresponding row. Upward arrows show phases that lag behind the benchmark phase while downward arrows show phases that lead the benchmark phase.

by CLaMS-ERA. This discrepancy may be related to lower values of $H_2O_{CH_4}$ and $\alpha$ in the upper stratosphere as shown in Fig. 2(A3), which result from a systematically faster stratospheric circulation in ERA-I.

The amplitude and phase of the QBO signal in SWV show pronounced uncertainty in the middle stratosphere (dashed square region in Figs. 9 and 10). This region coincides with the strongest QBO signal in zonal wind. During the MLS period (2005–

5   2013), both CLaMS-ERA and CLaMS-JRA overestimate $A_{\mathrm{QBO}}$ in this region, particularly within the 0-20°N latitude band. These overestimates can be attributed to strong upward transport of $H_2O_e$ from the lower stratosphere. Although CLaMS-MRA underestimates $A_{\mathrm{QBO}}$ in this region, the spatial pattern produced by CLaMS-MRA in the middle stratosphere is more consistent with that indicated by MLS. The estimate of phase in this region from CLaMS-MRA shows ∼6 months difference when years before MLS period are included or excluded (see Fig. 10D1-D2, where phase shifts are indicated by arrows).

10  Uncertainties in QBO-related $H_2O$ anomalies arise not only from different circulation responses but also from disagreements in zonal wind at the equator among the reanalyses. Equatorial winds at 10 hPa (∼800–900 K) in MERRA-2 are clearly different from the FUB records, ERA-I, and JRA-55 during the 1980s and early 1990s, when the assimilated observations are less able to pull the forecast model away from its own internally-generated QBO signal (Coy et al., 2016; Kawatani et al., 2016). The four estimates are more consistent at 30–50 hPa (∼500–600 K).




## 6   Trends and residual sources of natural variability

As mentioned in section 2.3 and observed in the SWOOSH record (see bottom panel of Fig.1), residual variability beyond the AC, QBO and SAO ($\chi_{\mathrm{res}}(t)$ in eq.(4)) also contributes substantially to variations in $H_2O$ entry mixing ratios. Figure 11A

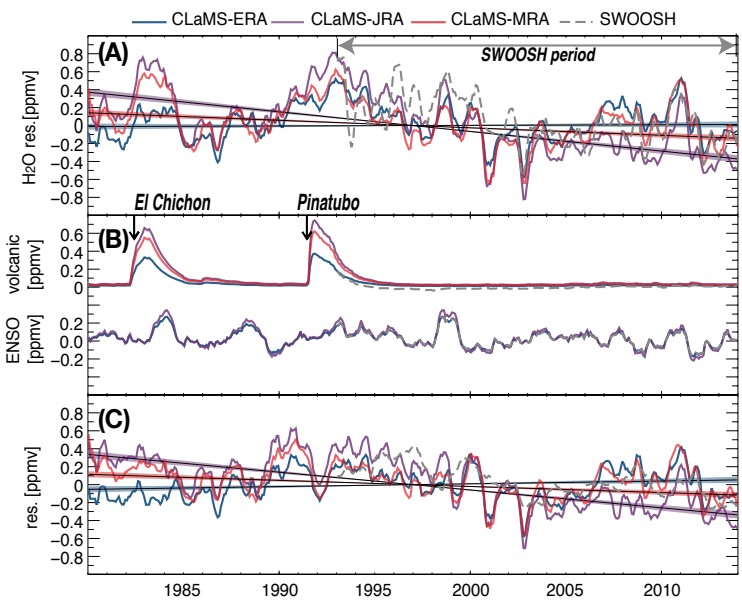

**Figure 11.** Comparison of residual variability after removing the AC, QBO and SAO from $H_2O$ entry values based on CLaMS simulations and SWOOSH. Panel A shows the residual term ($\chi_{\mathrm{res}}$) after applying a 3-month running mean. The long-term linear fit is plotted as a solid line with $\pm 2\sigma$ uncertainties. Panel B shows additional components of the multiple linear regression, including the influences of variations in volcanic aerosol (upper set of lines) and ENSO (lower set of lines). Panel C shows the residuum of A after subtracting all contributors shown in B, along with the corresponding trend.

shows that the three CLaMS simulations produce similar interannual variability in the $\chi_{\mathrm{res}}(t)$ term. Within the SWOOSH
5  period (1993–2013), representations of interannual variability among the simulations and the SWOOSH analysis are more consistent during the Aura MLS period (2004–2013) than the pre-MLS period (1993–2004), likely due to contemporaneous improvements in the quantity and quality of satellite observations assimilated by the reanalysis systems (Fujiwara et al., 2017). However, significant differences are evident in the simulated $H_2O$ entry mixing ratio trends, as discussed later in this section.

Among the additional sources of climate variability, ENSO and stratospheric volcanic aerosols have been shown to modulate
10  both the tropical ascending branch of the BDC (e.g. Diallo et al., 2017, 2018a, b) and tropical tropopause temperatures (e.g. Holton and Gettelman, 2001; Mitchell et al., 2015; Hu et al., 2016). Consequently, variations in ENSO and stratospheric volcanic aerosols have significant influences on $H_2O$ entry mixing ratios (e.g. Scaife et al., 2003; Fueglistaler and Haynes, 2005; Konopka et al., 2016; Diallo et al., 2017). To clarify the ENSO and volcanic aerosol impacts on the $H_2O$ entry mixing ratios



in each simulation, we analyze the residuum of $H_2O$ entry values ($\chi_{\mathrm{res}}(t)$ in eq.(4)) using the following multiple regression model:

$$\chi_{\mathrm{res}}(t) = a * P_{\mathrm{volcano}}(t - \tau_{\mathrm{volcano}}) + b * P_{\mathrm{ENSO}}(t - \tau_{\mathrm{enso}}) + \chi'_{\mathrm{res}}(t), \qquad (5)$$

where $P_{\mathrm{ENSO}}$ is the normalized Multivariate ENSO Index (MEI; (Wolter and Timlin, 2011)) and $P_{\mathrm{volcano}}$ is the aerosol optical depth (AOD) as recorded in satellite data (Vernier et al., 2011). The coefficients are the amplitude $a$ and lag $\tau_{\mathrm{volcano}}$ associated

with volcanic aerosols and the amplitude $b$ and lag $\tau_{\mathrm{enso}}$ associated with ENSO. We determine the parameters $a$, $b$, $\tau_{\mathrm{volcano}}$ and $\tau_{\mathrm{enso}}$ as the parameter set that minimizes the residual ($\chi'_{\mathrm{res}}$) in the least-squares sense. Additional details of the method and its application have been summarized by Diallo et al. (2017).

Figure 11B shows the response of $H_2O$ to stratospheric volcanic aerosol (upper set in panel B) and ENSO (lower set in panel B) based on the simulations and the observations. Beyond some differences in magnitude, the volcanic aerosol-induced

changes in $H_2O$ entry values agree well in terms of response signs and lags. The increase in modelled $H_2O$ associated with volcanic aerosols arises mainly from aerosol-induced warming of the TTL (Mitchell et al., 2015; Diallo et al., 2017). Positive $H_2O$ entry anomalies induced by the two major volcanic eruptions during the CLaMS period are twice as large in CLaMS-JRA and CLaMS-MRA (0.6 ppmv for El Chichón and 0.8 ppmv for Pinatubo) as in CLaMS-ERA (0.3 ppmv for El Chichón and 0.4 ppmv for Pinatubo). Note that among these three reanalyses, only MERRA-2 explicitly includes perturbations to the

stratospheric aerosol burden following volcanic eruptions (Fujiwara et al., 2017), although the effects of these perturbations may also enter all of the reanalyses indirectly through the assimilation of temperature, ozone and other affected quantities.

As with the volcanic aerosol-induced effects, ENSO impacts on $H_2O$ entry values from CLaMS and SWOOSH agree well with respect to the main characteristics of the response. All four estimates show positive $H_2O$ entry anomalies during El Niño and negative anomalies during La Niña, consistent with previous studies (e.g. Randel et al., 2009; Calvo et al., 2010; Konopka

et al., 2016; Diallo et al., 2018b). Lags ($\tau_{\mathrm{enso}}$) in ENSO-induced $H_2O$ entry anomalies relative to the ENSO signal (here represented by the MEI index) are likewise consistently around one year among all three simulations and the observations. However, there are notable quantitative differences in the magnitude of $H_2O$ entry mixing ratio changes induced by ENSO variability. CLaMS-JRA and CLaMS-MRA show increases in $H_2O$ entry mixing ratios about $\sim 0.1$ ppmv larger than CLaMS-ERA during strong El-Niño events (1982/1983, 1986/1987 and 1997/1998). However, the response in the latter exhibits the

best agreement with SWOOSH during the 1997/1998 El Niño.

Figure 11C shows the residual terms ($\chi'_{\mathrm{res}}$) and the trends related to the residual terms after removing ENSO and volcanic aerosol-induced variability. We notice that the significant differences among the trend estimates shown in Fig. 11A largely remain in the residual variability. Further quantification of contributions to the trends is listed in Table 1. We also notice a substantial quasi-decadal variability remaining in the residuals. After removing the high-frequency variability and the linear

trends, this quasi-decadal variability agrees well among the three simulations as well as with the SWOOSH data. Further interpretation of this consistent quasi-decadal signal is out of the scope of this study that aims to intercomparison of different reanalyses.





**Table 1.** Trend estimates in $H_2O$ entry mixing ratios and contributions from different sources of variability based on the three CLaMS simulations and SWOOSH over the SWOOSH period (1993–2013) and the longer CLaMS period (1980–2013).Trends are reported in units of ppmv decade$^{-1}$ (left of the slash) and % decade$^{-1}$ (right of the slash).

|  | SWOOSH | CLaMS-ERA | CLaMS-JRA | CLaMS-MRA |
|---|---|---|---|---|
| $C_{trd}$ |  |  |  |  |
| **1993–2013** | -0.22/-5.8 | -0.06/-1.6 | -0.39/-8.6 | -0.06/-1.5 |
| **1980–2013** |  | 0.01/0.2 | -0.22/-4.9 | -0.08/-2.0 |
| $\Delta C_{trd}$(**ENSO**) |  |  |  |  |
| **1993–2013** | -0.05/-1.3 | -0.06/-1.5 | -0.07/-1.6 | -0.05/-1.3 |
| **1980–2013** |  | -0.02/-0.6 | -0.03/-0.6 | -0.02/-0.5 |
| $\Delta C_{trd}$(**volcano**) |  |  |  |  |
| **1993–2013** | 0.00/0.0 | 0.02/0.4 | 0.03/0.7 | 0.03/0.6 |
| **1980–2013** |  | 0.00/0.0 | 0.00/0.0 | 0.00/0.0 |
| $\Delta C_{trd}$(**res**) |  |  |  |  |
| **1993–2013** | -0.18/-4.6 | -0.02/-0.6 | -0.35/-7.8 | -0.03/-0.8 |
| **1980–2013** |  | 0.03/0.7 | -0.19/-4.3 | -0.06/-1.5 |

Table 1 lists the trends in $H_2O$ entry mixing ratios along with the estimated contributions of ENSO and volcanic aerosols to trends in the three reanalyses-driven simulations and SWOOSH over the shorter period (1993–2013) and the longer-term period (1980–2013). Trends are calculated by analyzing the residual of the $H_2O$ entry anomalies with and without the ENSO and volcanic aerosol signals. The trend produced by CLaMS-ERA is very consistent with that produced by CLaMS-MRA after

1985, as is the residual variability shown in Fig. 11C. By contrast, CLaMS-JRA produces a strong negative trend. As non-periodic variability, such as ENSO variability and the volcanic aerosol burden, may contribute to these trends, their estimated contributions are listed in the second and third rows of Table 1. ENSO variability contributes to negative trends in both modelled and observed $H_2O$ entry mixing ratios over the 1993–2013 SWOOSH period. For the entire simulation period (1980–2013), the ENSO contribution to trends is also negative but with a reduced magnitude. The variability due to volcanic eruptions has zero-

trend contribution to the $H_2O$ entry values over the longer period (1980–2013), which is consistent with the volcanic-aerosol induced zero-effect on the trends of mean age of air at the tropical lower stratosphere shown in Diallo et al. (2017). However, it has a positive trend contribution during the 1993–2013 SWOOSH period, likely due to the increased frequency of minor volcanic eruptions after 2008 (Diallo et al., 2017). Conversely, the volcanic aerosol contribution to the trend from 1980 to 2008 (not shown) is negative due to the forcing of the two major eruptions. This analysis suggests that the contribution of volcanic

aerosols to interannual variability in $H_2O$ entry values is highly period-dependent. Although we find near compensation over the CLaMS period used in this work (1980–2013), trends in SWV can be substantially influenced by periods of enhanced or reduced supplies of volcanic aerosol to the stratosphere.



Differences among the trend estimates persist in the residual variability even after accounting for the effects of ENSO and volcanic aerosol (Fig. 11C and last row in Table 1), indicating that uncertainties in the model-based trends do not emerge from different responses to major volcanic eruptions or strong ENSO events. The model-based $H_2O$ trends are sensitive to the representation of the tropical tropopause temperature from reanalyses, especially for the early years of the satellite era (e.g.

5   1980–1985) (Fueglistaler et al., 2013). Differences in the observations assimilated by the three reanalyses or in the representation of the BDC may also contribute to differences in the trend estimates. Reliable attribution of differences in simulated SWV trends will require disentangling complex interactions among changes in tropopause temperature, the stratospheric circulation, and anthropogenic factors such as the amount of methane entering the stratosphere at global scale.

## 7   Discussion

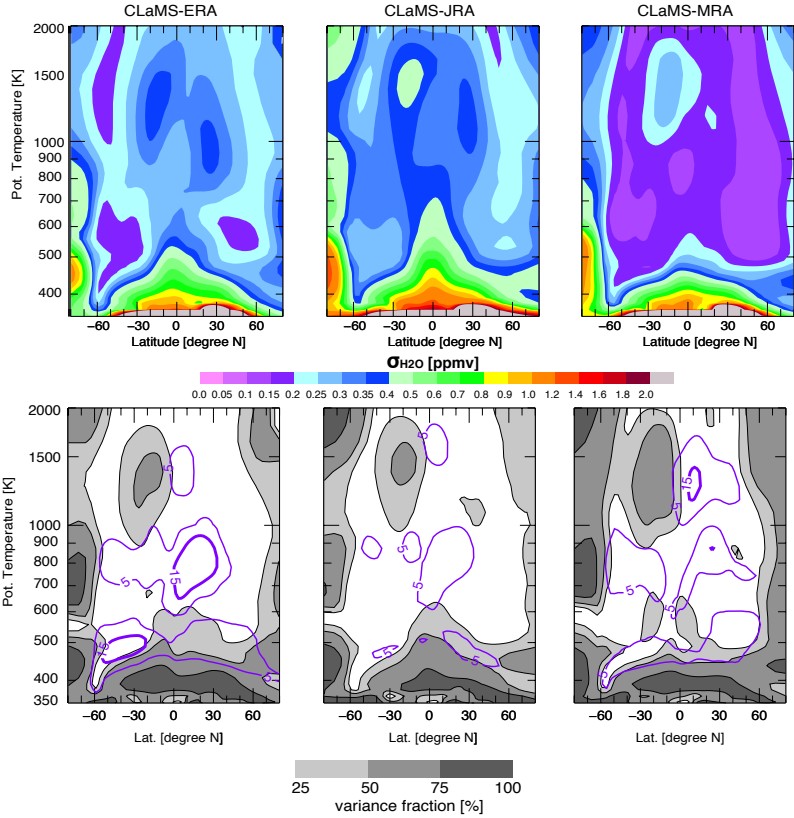

**Figure 12.** Total variance in monthly-mean $H_2O$ ($\sigma_{H_2O}$; upper row) and the fraction of periodic variance relative to total variance (in %; bottom row) for CLaMS-ERA (left), CLaMS-JRA (center) and CLaMS-MRA (right). Shading in the lower panels shows the variance fraction attributed to the annual cycle, while purple contours show the variance fraction attributed to the QBO.



In the sections above, we relate differences in representations of the climatological mean, AC and QBO signals of SWV to differences in upwelling rates in the shallow branch of the BDC among the reanalysis products used to drive the model. One remaining question is whether these differences in upwelling strength are also responsible for differences in AC or QBO variance. Analysis of the total variance and its contributors in each simulation can shed light on this question. If different up-

welling strengths are the main factor, then these different rates of vertical advection should influence each contributor similarly to the total variance. In other words, a larger (or smaller) magnitude of the AC or QBO variance should be approximately in proportion to a larger (or smaller) magnitude of the total variance. We use the standard deviation from the full $H_2O$ time series ($\sigma_\chi$, where $\chi$ is the $H_2O$ volume mixing ratio) to quantify the total variance. This metric is shown for each of the three simulations in the upper row of Figure 12. We then check the fraction of the total variance attributable to the AC and QBO,

respectively: $\sigma_{\chi AC}/\sigma_\chi$ and $\sigma_{\chi QBO}/\sigma_\chi$. The variance fractions attributable to the AC and QBO components of the total signal are shown in the bottom row of Figure 12 as grey shading and purple contours, respectively.

Figure 12 indicates that zonal-mean distributions of AC and QBO variance fractions are very consistent across all three simulations (bottom row), although the magnitudes of total variance differ to some extent among the three CLaMS runs (top row). For example, CLaMS-MRA produces the smallest amplitudes of both the AC and QBO signals in the lower and middle

stratosphere among the three simulations. The top right panel of Fig. 12 (corresponding to CLaMS-MRA) shows that these smaller amplitudes are in turn associated with weaker total variance in the middle stratosphere. By contrast, the fractional contributions attributed to the AC and QBO (bottom right panel) are comparable to if not larger than the corresponding fractional contributions based on CLaMS-ERA or CLaMS-JRA. This similarity among variance fractions implies that differences in the strength of the tropical upwelling introduced via the prescribed reanalysis diabatic heating rates can adequately explain

differences in the amplitudes of the AC and QBO signals produced by the CLaMS simulations.

The reason that diabatic heating rates in the tropical lower stratosphere are smaller in MERRA-2 than in ERA-I or JRA-55 is as yet unclear. Although a full attribution is beyond the scope of this study, we consider here two possible contributors that may be informative for other users of these data products: 1) the larger long-wave cloud radiative effect (LWCRE) in MERRA-2 than ERA-I or JRA-55; and 2) the unique assimilation process used in MERRA-2.

Diabatic heating rates immediately above the tropical tropopause are dominated by radiative heating $Q$ (e.g. Wright and Fueglistaler, 2013), which, under the plane parallel assumption applied in the reanalysis models, indicates the net vertical convergence of energy in the form of radiation. Adopting the Newtonian cooling approximation $Q \sim -\alpha(T - T_{eq})$ (with $\alpha$ the inverse of the radiative relaxation time), radiative heating rates $Q$ can be treated as inversely proportional to the difference between the local temperature $T$ and a local radiative equilibrium temperature $T_{eq}$ (e.g. Fueglistaler et al., 2009a). The latter

depends on the vertical profile of temperature, the chemical composition, and the radiative effects of aerosol and clouds within the atmospheric column. Figure 13 illustrates the relationship between daily-mean gridded diabatic heating rates in the tropical lower stratosphere $\dot{\theta}_{LS}$ (30°S–30°N; $\theta = 380$ K to $\theta = 460$ K) and the corresponding LWCRE (defined as clear-sky minus all-sky outgoing long-wave radiation at the nominal top of atmosphere) based on ERA-I, JRA-55, and MERRA-2. Daily-mean diabatic heating rates and associated upwelling in the tropical lower stratosphere are negatively correlated with LWCRE in all

three reanalyses, with $r$ ranging from $-0.18$ in JRA-55 to $-0.38$ in ERA-I and $-0.54$ in MERRA-2. This negative relationship





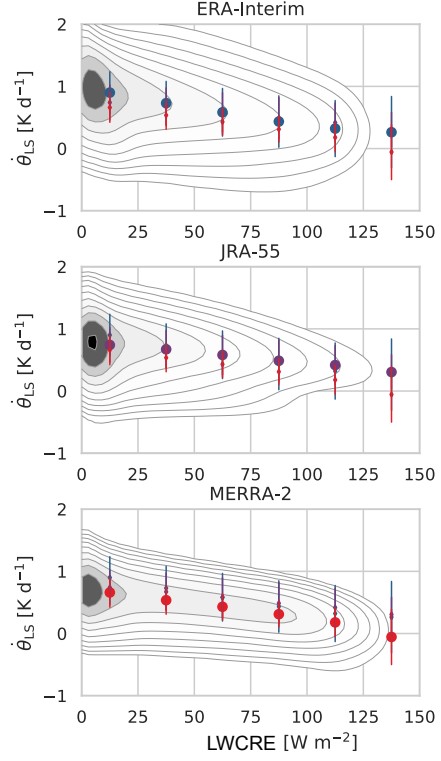

**Figure 13.** Joint distributions of daily-mean gridded diabatic heating rates ($\dot{\theta}$) in the tropical lower stratosphere ($\theta = 380\,\text{K}$ to $\theta = 460\,\text{K}$) against the corresponding long-wave cloud radiative effect (LWCRE) based on ERA-I (upper), JRA-55 (center), and MERRA-2 (lower) during 1980–2013. The LWCRE is calculated as clear-sky minus all-sky outgoing long-wave radiation. Also shown are means and standard deviations of $\dot{\theta}_{\text{LS}}$ composited into LWCRE bins in intervals of $25\,\text{W}\,\text{m}^{-2}$ starting from zero. Estimates based on ERA-I (blue), JRA-55 (purple), and MERRA-2 (red) are shown on all three panels to facilitate comparison. Distributions are based on variables archived on $1°\times 1°$ (ERA-I) and $1.25°\times 1.25°$ (JRA-55 and MERRA-2) latitude–longitude grids between $30°\text{S}$ and $30°\text{N}$.

may be explained by noting that clouds occur almost exclusively within the troposphere, so that a larger LWCRE corresponds to a smaller upward flux of long-wave radiation across the tropopause. All else remaining equal, a reduced upward flux across the tropopause acts to reduce $T_{\text{eq}}$ and the net convergence of radiant energy, and therefore implies a smaller diabatic heating rate. Moreover, values of the LWCRE in MERRA-2 (median value: $15.9\,\text{W}\,\text{m}^{-2}$) are systematically larger than those in ERA-I
5   ($10.0\,\text{W}\,\text{m}^{-2}$) or JRA-55 ($6.8\,\text{W}\,\text{m}^{-2}$), especially at the upper end of the range (cf. $90^{\text{th}}$ percentile values of 82.9, 50.1, and $34.8\,\text{W}\,\text{m}^{-2}$, respectively). It is thus unsurprising that MERRA-2 produces weaker diabatic upwelling near the base of the tropical pipe. However, this effect does not appear to account for the entire difference, as lower stratospheric heating rates composited for the same ranges of LWCRE are still systematically smaller in MERRA-2 than in ERA-Interim or JRA-55 (Fig. 13).




Another potential contributor is the unique assimilation process used in MERRA-2 (and its predecessor MERRA), which includes an additional 'corrector' step after the initial 3D-Var analysis (i.e. an 'iterative predictor-corrector approach'; see also Bloom et al., 1996; Rienecker et al., 2011; Fujiwara et al., 2017). During the corrector step, all analysis increments are applied gradually over time as additional tendency terms. This procedure improves the internal consistency among analyzed variables

and other model variables. However, it also means that the diabatic tendency and other physical tendency terms in MERRA-2 are archived during the corrector step, in combination with the analysis increment, whereas other reanalyses produce these terms during an initial forecast analogous to the predictor step in MERRA-2. This may have the unintended effect of systematically damping (or amplifying) the tendency terms produced by the physical parameterizations. Again adopting the Newtonian cooling approximation, systematic biases in $T$ or radiatively active constituents could result in the assimilation increment con-

stantly acting to reduce $T - T_{\mathrm{eq}}$ in the lower stratosphere, thereby damping radiative heating rates and associated diabatic upwelling. Using 'replay' simulations, which mimic the corrector forecast using a modified version of the atmospheric model and data assimilation system used for MERRA-2, Orbe et al. (2017) showed that a simulation constrained by time-averaged assimilated fields produced approximately 30% slower ascent in the tropical lower stratosphere than one based on instantaneous analysis fields. Although Orbe et al. (2017) concluded that the slower ascent in the simulation based on time-averaged

assimilated fields produced output in better agreement with available observations, their result nonetheless highlights the potential impact of the predictor–corrector approach on upwelling in the tropical lower stratosphere. Additional evidence that data assimilation may suppress lower stratospheric upwelling in MERRA-2 is provided by comparison against the MERRA-2 AMIP dataset (Collow et al., 2017), a ten-member ensemble of free-running simulations generated using the same model and boundary conditions as MERRA-2 but without data assimilation. Time-mean zonal-mean diabatic heating rates in the tropical

lower stratosphere are about $0.1 \sim 0.2\,\mathrm{K\,day^{-1}}$ smaller in MERRA-2 than in MERRA-2 AMIP over the period 1980–2017 (see Figure D1 in Appendix D).

## 8   Conclusions

We evaluate representations of SWV and its variations at multiple timescales using the Chemical Lagrangian Model of the Stratosphere (CLaMS) driven by horizontal winds and diabatic heating rates from three recent atmospheric reanalyses: ERA-I,

JRA-55 and MERRA-2. The analysis is based on CLaMS simulations of monthly-mean zonal-mean $H_2O$ from 1980–2013. We present an intercomparison of simulated variations in $H_2O$ in the context of observational estimates from Aura MLS and the Stratospheric Water and Ozone Satellite Homogenized (SWOOSH) database, focusing on the annual cycle (AC), the quasi-biennial oscillation (QBO), and long-term variability and trends. Based on the results of this intercomparison, we reach the following conclusions.

The climatological means of SWV, that represents a combination of $H_2O$ entering the stratosphere through the tropical tropopause (dominant in the lower stratosphere) and $H_2O$ supplied by $CH_4$ oxidation (dominant in the upper stratosphere), are in a good agreement (within $\pm 10\%$ relative differences) among the three simulations. The age-of-air and the ratio of $H_2O$ from $CH_4$ oxidation in the lower stratosphere are larger in CLaMS-MRA than in the other two runs, consistent with relatively weaker




diabatic upwelling in the lower and middle tropical stratosphere in MERRA-2 relative to ERA-I or JRA-55. Mean tropopause temperatures in ERA-I are approximately 1 K colder than those in JRA-55, resulting in a tropical lower stratosphere that is ∼0.6 ppmv drier in CLaMS-ERA than in CLaMS-JRA. CLaMS-MRA produces a moderate $H_2O$ entry mixing ratios close to observed as a result of moderate mean tropopause temperature among the three reanalyses.

Differences in amplitudes of modelled $H_2O$ entry mixing ratio ACs can largely be explained by differences in mean tropical tropopause temperatures. CLaMS-MRA produces an AC phase close to observed while CLaMS-ERA and CLaMS-JRA both produce annual maxima one month earlier than observed. Tuning of the dehydration scheme could possibly eliminate differences in both mean entry values and the AC amplitude, but not the phase (see also Liu et al., 2010; Fueglistaler et al., 2013). The spatial distribution of tropopause temperatures is also important to understanding differences in seasonal $H_2O$ entry mix-

ing ratios. The relative dryness of CLaMS-ERA is most pronounced in boreal winter, consistent with especially large negative biases in tropopause temperatures over the western Pacific in ERA-I relative to MERRA-2. By contrast, CLaMS-JRA produces particularly large values of $H_2O$ entry mixing ratios during boreal summer, consistent with systematically warmer tropopause temperatures over the Bay of Bengal and tropical Pacific.

     The AC and QBO signals in simulated SWV are robust across the three simulations and reasonably consistent with ob-

servations. The main discrepancies in the AC and QBO components of $H_2O$ variability are located in the tropical lower and middle stratosphere. The observed AC and QBO signals are generally located between those produced by CLaMS-ERA and CLaMS-JRA (too strong) and those produced by CLaMS-MRA (too weak). That is mainly linked to the fact that the upwelling rates in the tropical lower stratosphere are smaller in MERRA-2 than ERA-I or JRA-55, which is attributable to the larger long-wave cloud radiative effect and the unique assimilation process in MERRA-2. CLaMS-MRA produces more realistic

pattern of QBO-related signals in SWV (but a bit weaker than observation) than the other two runs, especially in the middle stratosphere. The 'tape recorder' signal is 25% weaker and ∼1.5 month slower (between 450 K and 600 K) in CLaMS-MRA relative to SWOOSH. By contrast, the tape-recorder signals are both stronger and faster than observed in CLaMS-ERA (20% stronger and ∼1 month faster than SWOOSH) and CLaMS-JRA (40% stronger and ∼1 month faster than SWOOSH). We find that differences in the rate of tropical upwelling among the reanalyses not only modulate the propagation of the simulated

tape-recorder signal, but also interact with $CH_4$ photochemistry in ways that tend to amplify the differences due to propagation rates alone (section4.1).

     With respect to residual variability in SWV, the consistency among the model results and the SWOOSH analysis improves with time, including the responses of SWV to variations in ENSO and volcanic aerosol. All simulations indicates strong and consistent quasi-decadal variability. Trends in $H_2O$ entry mixing ratios over the 1980–2013 period are highly sensitive to the

reanalysis used to drive the model, leading to different magnitudes and even different signs. This sensitivity of simulated $H_2O$ trends to choice of reanalysis cannot be fully explained by discrepancies in the response of modelled $H_2O$ to ENSO variability or major volcanic eruptions. The $H_2O$ trends are particularly sensitive to the quality of the reanalyses during the first five years of the analysis period (1980–1985).

     As best estimates of the true state of the atmosphere, meteorological reanalyses are widely used to drive tracer transport

or constrain the meteorological state during model simulations. Our study indicates that the ERA-I, JRA-55, and MERRA-



reanalyses are all capable of reproducing the seasonality and interannual variability of SWV related to QBO, ENSO and volcanic aerosol. However, particular attention should be paid to potential reanalysis-dependent biases when studying longer-term variability in SWV or other atmospheric tracers. Moreover, the available observations allow no conclusions about which reanalysis is most realistic. Therefore, multiple reanalyses should be used whenever possible to better characterize the effects
of uncertainties in the atmospheric state.

## Appendix A: SWOOSH-period of H$_2$O data used for comparison with CLaMS

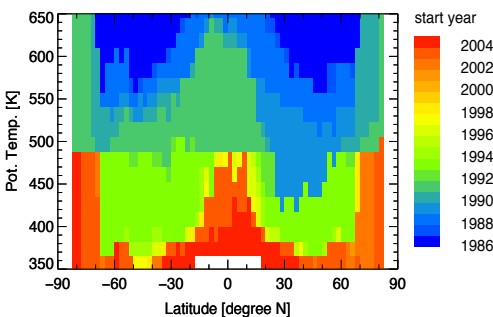

**Figure A1.** Distribution of start years for the 'SWOOSH period' used to compare against the model results. The start time for each latitude-$\theta$ grid point is the first month for which more than 12 months of SWOOSH H$_2$O data are available over the following 2 yr.

In section 2.2, we have clarified the procedure used to determine the 'SWOOSH period'. The start time for each latitude-$\theta$ grid location is the first month when SWOOSH H$_2$O has more than 12 months of available datawithin the n ext 2 years. The end time is always December 2013. This procedure excludes some early months of the SWOOSH record (mainly intermittent
observations from SAGE-II in the lower stratosphere), thereby improving the continuity of the H$_2$O time series. Figure A1 shows the distribution of 'SWOOSH period' start years identified via this procedure. Due to data filtering applied during the production of SWOOSH (Davis et al., 2016), the 'SWOOSH period' is typically: 1) shorter at lower levels than at higher levels; 2) shorter in the tropics and polar regions than in the mid-latitudes; 3) shorter in the Southern Hemisphere than in the Northern Hemisphere. The shortest 'SWOOSH periods' start from 2004, and are found in the tropical lower stratosphere and
at Southern Hemisphere high latitudes. The longest 'SWOOSH period' starts from 1986, in the mid-latitudes between $\theta$=550 K and $\theta$=650 K.

Although SWOOSH H$_2$O data has some gaps from the start month (shown in Fig. A1) to end of 2013, the full monthly and zonaly mean modelled H$_2$O are taken into account for comparison. This allows us to keep the modelled H$_2$O time series as continuous as possible, which in turn allows us to extract more reliable variance estimates. The results are virtually unchanged
when (SWOOSH) missing months are also excluded from the CLaMS runs.





## Appendix B: Tropospheric methane in CLaMS

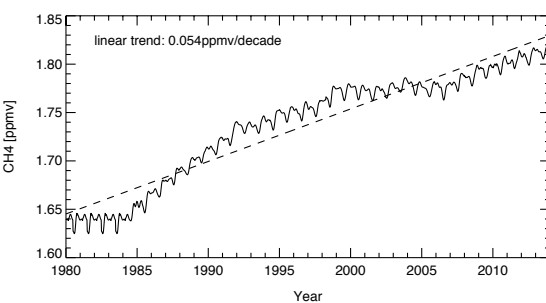

**Figure B1.** Temporal variations from 1980–2013 in the global mean methane mixing ratio prescribed at the model lower boundary, i.e. the hybrid vertical coordinate $\zeta = 0$-$100$ K.

The boundary condition for $CH_4$ mixing ratios in CLaMS is prescribed in the lowest model layer, corresponding to the hybrid vertical coordinate $\zeta = 0$-$100$ K. For the simulation period between 1985 and 2012, $CH_4$ measurements from the NOAA/CMDL ground-based measurement network are used (Masarie and Tans, 1995; Novelli et al., 2003). For simulations in 2012 and later,

observations from the AIRS instrument are used. The $CH_4$ mixing ratios between 1980 and 1984 in Fig. B1 are just repetition of the year 1985 since the NOAA/CMDL data is not available before 1985. As $CH_4$ is a long-lived tracer, we use zonal mean mixing ratios to prescribe the lower boundary condition. Figure B1 shows the time series of global-mean $CH_4$ mixing ratios prescribed at the model lower boundary. Since the time series before 1985 is artificial, Fig. B1 shows a start of the methane from the mid-1980s before leveling off in the mid-1990s. The long-term trend in $CH_4$ at the model lower boundary (1980–

2013) is $0.054$ ppmv decade$^{-1}$. In fact, there is one option to extend the period of tropospheric methane before 1985 following Fueglistaler and Haynes (2005) by using the Antarctic ice core data (Etheridge et al., 1992). We consider this option for the lower boundary condition in the future CLaMS development.

## Appendix C: Semi-Annual Oscillation (SAO)

Figure C shows the spatial distribution of the amplitude and corresponding variance fraction of Semi-Annual Oscillation (SAO)

signals in simulated SWV. The enhanced SAO signal in region 1 is associated with the semi-annual oscillation of the upper stratospheric circulation. Satellite observations of the SAO feature in the tropical upper stratosphere and the mechanisms behind this feature have been studied by Jackson et al. (1998), Randel et al. (1998), and Jackson et al. (1998). The enhanced SAO signals in region 2 and region 3 are located between two strong AC signals in the polar regions (SH > NH; see Fig. 7). The mechanism behind the SAO signals in regions 2 and 3 is related to the combined effects of seasonality in the formation of PSCs

and seasonality in vertical transport (e.g. Lossow et al., 2017b).





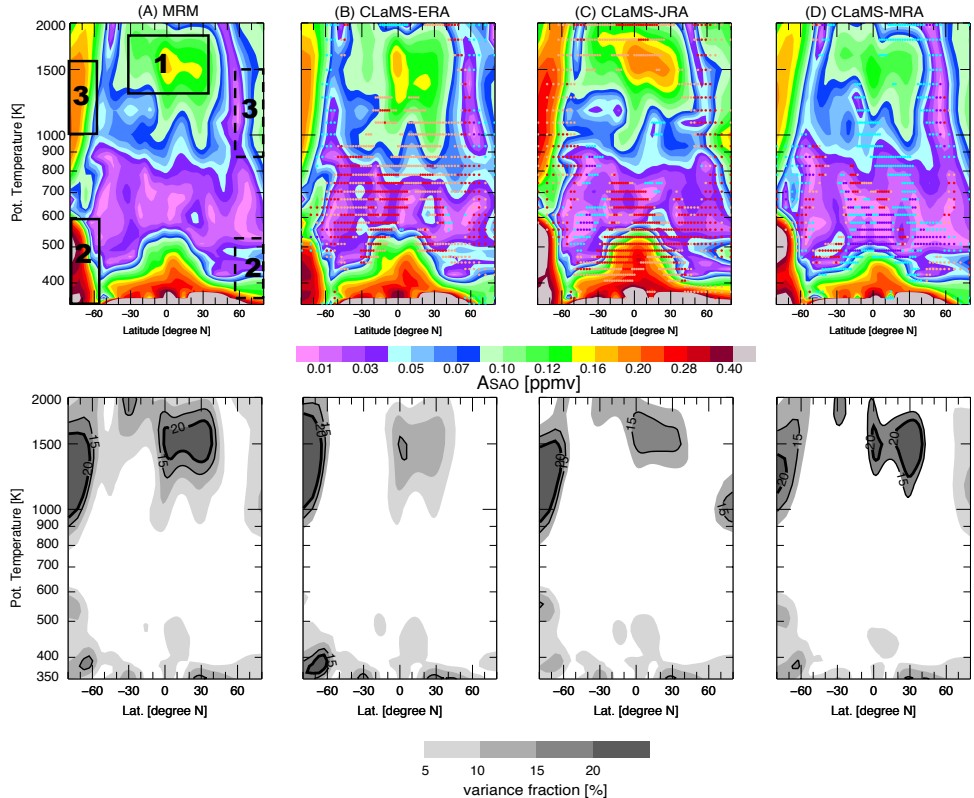

**Figure C1.** The top row shows SAO amplitudes in the same way as in the bottom row of Figure7. The bottom row shows the fraction of SAO variance relative to total variance, similar to the bottom row of Figure 11.

The SAO fraction shows that the SAO contributed more than 10% of the total variance in region 1 and region 3 (both located in the upper stratosphere). The SAO amplitudes in region 1 simulated by the CLaMS runs are in good agreement (within 0.1-0.2 ppmv), and also agree well with HALOE and MLS observations (Jackson et al., 1998; Lossow et al., 2017b). As with the AC and QBO features, the SAO signal in region 1 is most pronounced in CLaMS-JRA, followed by CLaMS-ERA and CLaMS-

5  MRA. Similarly, the SAO fraction relative to the total variance is again higher in CLaMS-MRA (greater than 15–20%) due to the low total variance simulated by CLaMS-MRA in the mid-upper stratosphere.

## Appendix D: MERRA-2 versus MERRA-2 AMIP diabatic heating rates

The MERRA-2 AMIP dataset is an ensemble of ten Atmospheric Model Intercomparison Project (AMIP)-type simulations conducted using the atmospheric model used to produce MERRA-2. All simulations used identical boundary conditions and

10  model configurations to MERRA-2 but did not assimilate observations (Collow et al., 2017). Therefore, a comparison of



MERRA-2 against that of MERRA-2 AMIP gives a rough indication of the effect of the iterative predictor–corrector data assimilation procedure used in MERRA-2. The comparison of the diabatic heating rates is shown in Figure D1. The climato-

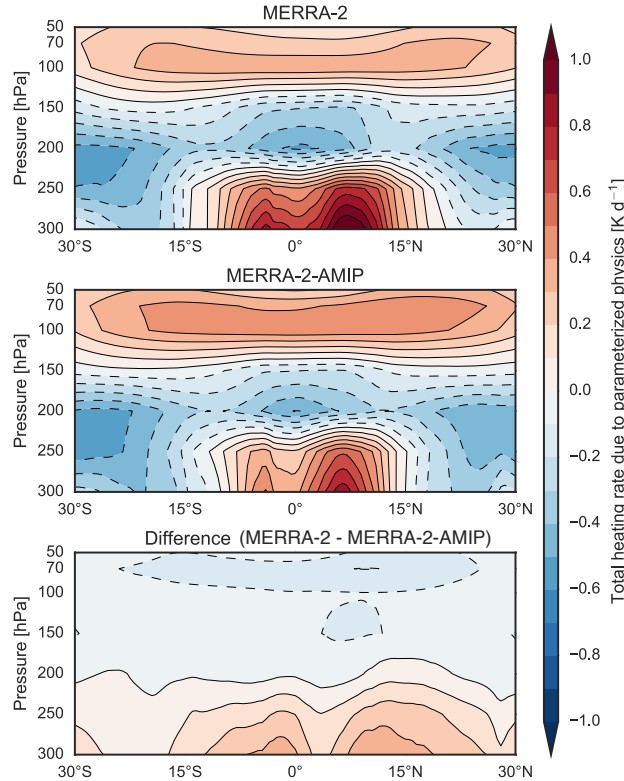

**Figure D1.** Comparison of time-mean zonal-mean diabatic heating rates in the tropical (30°N-30°S) lower stratosphere between the MERRA-2 reanalysis (top panel) and the MERRA-2 AMIP ensemble mean (middle panel) over 1980–2017. Differences between MERRA-2 and MERRA-2 AMIP are shown in the bottom panel.

logical patterns in MERRA-2 and MERRA-2 AMIP are qualitatively similar; however, their quantitative differences are substantial (bottom panel). Most notably, diabatic heating rates are about $0.2\sim0.4$ K day$^{-1}$ larger in MERRA-2 than in MERRA-2
5 AMIP at pressures greater than 200 hPa, while diabatic heating rates in the lower stratosphere (pressures less than 100 hPa) are $0.1\sim0.2$ K day$^{-1}$ smaller in MERRA-2 than in MERRA-2 AMIP. Thes results suggest that the data assimilation procedure suppresses upwelling in the tropical lower stratosphere by as much as 20–40%.

*Author contributions.* M. Tao carried out the analysis on the simulations and on the reanalysis data. P. Konopka and F. Ploeger contributed the code for analysis. J.S. Wright prepared the MERRA-2 reanalysis data and provided the attribution for slow upwelling rates in MERRA-
10 2 for the discussion section. P. Konopka, F. Ploeger and X. Yan performed the simulations driven by ERA-I, JRA-55 and MERRA-2, respectively. M. Diallo contributed the regression on natural variability. S. Fueglistaler and M. Riese gave helpful suggestions, especially



on the dehydration scheme in the model and the interpretation of trend. M. Tao wrote the manuscript. All the co-authors provided helpful discussions and comments on the manuscript.

*Acknowledgements.* This research was supported by a joint DFG-NSFC Research project with DFG project number 392169209 and NSFC project number 20171352419. Mengchu Tao thanks the German Helmholtz-Gemeinschaft within the Helmholtz-CAS joint research group (JRG) "Climatological impact of increasing anthropogenic emissions over Asia", enabling her research position in Institute of Energy and Climate Research, Stratosphere (IEK-7), Forschungszentrum in Jülich during which this work was carried out. We thank the Helmholtz Association under grant number VH-NG-1128 (Helmholtz-Hochschul-Nachwuchsforschergruppe), providing the research funding for the young investigator group in IEK-7 including the co-authors Felix Ploeger and Mohamadou Diallo.



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
