# Peer review of "Multi-timescale variations of modelled stratospheric water vapor derived from three modern reanalysis products"

_Atmospheric Chemistry and Physics, 2019_

## Referee Comment (RC1) · Anonymous Referee #1 · 3 Mar 2019

This paper uses a single model (CLaMS) to calculate stratospheric water vapour values on the basis of a simple dehydration scheme combined with a simple methane oxidation scheme. The model is driven by reanalysis winds and temperatures (winds to determine transport and temperature to specify dehydration). Three different reanalysis datasets (ERA-I, JRA-55 and MERRA-2) are used to drive the model and the objective of the paper to compare predictions of stratospheric water vapour concentrations that result from the use of each of the three datasets.

I think that this paper potentially makes a useful contribution. Reanalysis datasets are in wide use for many different types of calculations and it is important to have on record

different measures of the differences between the datasets in common use. There have been previous studies of the differences in transport characteristics between these datasets, but the calculation of stratospheric water vapour considered here on the one hand requires a particular combination of transport and temperature information and on the other is of significant general interest, because of the radiative and chemical importance of stratospheric water vapour and of the continuing challenges in measuring its concentrations and in establishing a long-term observational record.

However I do consider that the paper could be improved in various ways to make it as valuable as possible to researchers working in this general subject area.

p2 l21: 'water vapour values entering the stratosphere are determined by ... in the upper troposphere' - seems odd statement - previous papers (e.g. Liu et al 2010, Fig. 12) have suggested LDPs are distributed over a layer centred on about 90hPa. 'in the TTL' would be a more usual description.

p2 l22: 'Lagrangian models provide more accurate records of the temperature histories of air parcels compared to Eulerian models' – statement is true by definition (Eulerian models don't provide such records) or could be misread as 'provide a more accurate representation of transport' which is questionable - there are advantages and disadvantages to the two different formulations.

p3 l13-14: '30% too fast', '30% too slow' are based on the tape recorder considered over some a particular range of heights - give range explicitly.

p4 l8: You don't give sufficient information here to specify the dehydration scheme – e.g. the fall speed has to be combined with a length scale (specified in Ploeger et al 2013 as 300m). A first point is that if I look at the references you give – e.g. Poshyvailo et al (2018) I don't see sufficient information to be able to reproduce the scheme (e.g. that paper doesn't seem to say what the assumed fall speed it). My suggestion is that you provide all the necessary information together in an Appendix. A second point is that you mention various details such as sedimentation rate but then you fail to mention

the length scale - which is surely chosen on a completely ad hoc basis. So the naive reader might will infer that dehydration scheme is based on a precise physical model, whereas in fact it requires choice of length scale - which is surely ad hoc. (Ploeger et al 2013 note that it is comparable to the vertical resolution of the model - but that is hardly a physical justification.) So more clarity is needed - both for reproducibility and for an honest description of what is being done.

p4 l12: The information that greater intensity of small-scale mixing leads to moistening by ∼0.5ppmv and amplification of the annual cycle by ∼0.2 ppmv can't really be interpreted sensibly without more information. For example you could say that the range of mixing strength being considered is representative of actual uncertainty in small-scale mixing (which is what is said in the Poshyvailo et al 2018 paper).

p4 l14: 'best agreement with MLS observations' - I found this sentence a bit misleading. Of the papers cited, only Poshyvailo et al 2018 gives explicit information on the effect of varying mixing strength. So I don't see that the others provide any useful information on what choice of mixing strength gives the best agreement with MLS.

p4 l23: How are you calculating saturation mixing ratio exactly? Simmons et al (1999 QJRMS) make the point that the precise form of the expression used for saturation mixing ratio can be important. Again this is the sort of information that needs to be easily available to ensure that others can reproduce your calculations.

p4 l25: 'The effects of tuning ...' - this sentence would be clearer as 'The effects of tuning the critical supersaturation threshold in CLaMS have a similar effect to the effect of applying a frost point offset to the Lagrangian dry point temperature noted by Liu et al., (2010) and Fueglistaler et al., (2013), in that increase in the supersaturation threshold enhances both the mean value and the amplitude of the annual cycle in simulated H2O.'

p4 l28: I couldn't follow the logic of these two sentences - to me the point is that given uncertainty over the precise relation between temperature and saturation, e.g. due to

the uncertainty in the appropriate value of supersaturation threshold, it is impossible to be interpret differences between predicted water vapour and observations as being due to errors in LDP temperatures.

p4 l16: My experience is that the re-analyses don't provide diabatic heating as a single quantity, but as various components. Confirm that you are using all components (e.g. including latent heating).

p5 l5: My understanding is that you are estimating methane suppled water from (1), using a mode prediction of $CH_4\hat{}rec$. But how are you specifying alpha?

p6 l12-15: It seems important not to ignore the fact that the MLS weighting functions do not only produce 'artefacts' when applied to the CLaMS output, but that these artefacts may also be part of what is presented as the MLS observation - i.e. the MLS observations imply vertical structure that may be quite different to what is actually present. (At least that was my interpretation of Ploeger et al 2013.)

p6 l20: My understanding here is that you are simply extracting the semiannual cycle on the basis of its semi-annual frequency - not implying a direct relation to the phenomenon 'the Semi-Annual Oscillation (SAO)' that is identified in the low-latitude upper stratosphere. In other words, there is a potential difference in meaning between 'the SAO' and 'the semi-annual harmonic'.

p6 l26: It would be helpful here to note that the amplitudes and phases are what is presented in specific later figures.

p8 l21: It's a minor point, but you don't seem to have said explicitly that CLaMS-MRA is the MERRA simulation.

p12 l7: 'Moreover, this makes it possible that an optimisation of the dehydration scheme ...' - this sentence doesn't seem very relevant at this point in the text (and you have said something similar on p4).

p12 l14: 'However the AC phase of H2O entry values based on CLaMS-MRA is in better

agreement with the SWOOSH data' - given the uncertainty indicated in SWOOSH one wonders whether the better agreement is actually significant. The comment is mostly about September/October - i.e. the judgement all comes down to September-October differences. Whether or not CLaMS-MRA or ClaMS-JRA is in better agreement with SWOOSH during the J-F-M-A period is also open to question.

p12 l31: For me a strong part of the evidence for the importance of different regions are LDP distributions based on trajectory studies - again Fueglistaler et al (2005) given the relevant distributions.

p13 l6: Can the slow upwelling alone account for the seasonal signal in $H2O\_CH\_4$? - i.e. isn't the seasonal cycle in in-mixing at least as important?

p14 Figure 6: It would be helpful to have the propagation of the maximum values as well as the propagation of the minimum values explicitly marked on the Figures.

p15/16: Figures 7 and 8 - it would be much better if Figure 7 could be immediately followed by Figure 8 - perhaps they could even be combined into a single Figure. I found the overlaid dots in Figure 7 quite difficult to see and to interpret – of course when reading the paper on a screen one can zoom to look in more detail at interesting features, so one isn't applying the same rules of legibility that might have been applied for paper publications. But given that the dots represent relative error, the fact that some of them are the same colours as are used for the values of the depicted field is confusing. It might be clearer to do something like use overprinted + and - (of different size to indicate size of relative difference).

p17 l3: 'The phase of the QBO effect at the tropopause is therefore consistent with that of the 50hPa QBO wind' - the 'therefore' only makes sense if you say explicitly that it is the 50hPa wind (rather than the wind at some other level) that is correlated with the tropopause temperature.

p17 l6: 'both overestimate $A\_QBO$ at isentropic levels between 450K and 550K' ...

'both biases can be traced back to strong diabatic upwelling' - the key point surely is not necessarily that the diabatic upwelling is strong, but that the the circulation (upwelling + eddy mixing) is such that the tape recorder signal in these two analyses decreases in amplitude in the vertical slowly relative to MLS and MERRA (as you show for the annual cycle in Figure 6 and 7).

p17 Figure 9: Give information in the caption on the dashed box. You could call this 'region 3'. Incidentally this region doesn't really seem to be characterised by a 'clear peak' - contrary to what you say on p16 l14.

p17 l9: You say 'the large values of A_QBO in region 2 are mainly linked to QBO-related modulation of the stratospheric circulation.' You should provide a reference for this. For example this aspect of the QBO signal was considered in the Baldwin et al (2001) review. That was quoting previously published studies. There may have been more recent work on this topic.

p17 l11: This reference should be 'Lossow et al 2017b'. The 2017a reference doesn't mention the QBO at all.

p18 l3: 'The amplitude and phase of the QBO signal in SWV show pronounced uncertainty in the middle stratosphere' - what do you mean by this? Do you mean that the observational signal is uncertain? (I don't think so.) If you mean that there is disagreement between the different CLaMS simulations and between the CLaMS simulations and the re-analysis then say that directly.

p18 Figure 10: As previously noted for Figures 7 and 8, it would be helpful for the reader if Figures 9 and 10 were immediately adjacent (or combined into a single Figure).

p18 A key point about the QBO signal in water vapour is that it results in part from upward propagation of the water vapour variations 'imprinted' by the QBO temperature variation at the tropical tropopause and in part from QBO variations in the meridional circulation (which seem likely to be primarily to be responsible for the QBO signal in

water vapour seen in the upper stratosphere). You don't really exploit the distinction between these two processes generally in interpreting Figure 10 (which at first sight looks rather complicated). For example, you have identified from the annual cycle that ERA-I and JRA-55 seem to propagate signals too rapidly in the vertical in the lower stratosphere – presumably this means that their QBO signals are different in phase to the observed in the lower stratosphere, though you don't identify this as a significant difference (i.e. there is no vertical arrow marked corresponding to this in the lower stratosphere), presumably because the associated phase error is only a month or so. You do mark an arrow for JRA-55 - but don't comment on it. Do you interpret this as resulting from the strong vertical upwelling in JRA-55?

p18 l12: The fact that MERRA-2 doesn't match the observed QBO during the 1980s and early 1990s seems a pretty major and obvious problem to me. If that is the main cause of the difference in phase errors for MERRA between Figures D1 and D2 (indicated by the arrows in different directions) then it is probably clearer to omit D2 (and say why you have done that).

p26 l6: I've noted previously that the difference in annual cycle phase between predictions of different reanalyses seems to be judged almost on the basis of small differences between two months - and my question is whether that difference is really significant.

---

## Referee Comment (RC2) · Anonymous Referee #2 · 6 Mar 2019

Review of Tao et al 2019 "Multi-timescale variations of modelled stratospheric water vapor derived from three modern reanalysis products"

This paper evaluates the simulated stratospheric water vapor in three modern reanalyses in comparison with observations to better understand the inter-reanalysis differences. Overall, this paper is well written and well-supported in its arguments, and will be a nice contribution to the literature particularly for those wishing to better understand how the choice of input reanalysis for models (such as CTMs) will impact model results.

I have a few minor suggestions that should be considered:

[Figure]

Title/abstract: I believe through most of the paper the British English spelling of vapor (i.e., vapour) is used, but the American English spelling is used in the title and abstract. A consistent spelling should be used throughout the paper.

Line 17: There have been other attempts at merging observational records of WV. In particular, Froidevaux et al 2015.

Froidevaux, L., Anderson, J., Wang, H. J., Fuller, R. A., Schwartz, M. J., Santee, M. L., Livesey, N. J., Pumphrey, H. C., Bernath, P. F., Russell, J. M., III and Mc-Cormick, M. P.: Global OZone Chemistry And Related trace gas Data records for the Stratosphere (GOZCARDS): methodology and sample results with a focus on HCl, $H_2O$, and $O_3$, Atmospheric Chemistry and Physics, 15(18), 10471–10507, doi:10.5194/acp-15-10471-2015, 2015.

Lines 30-31: What versions of SWOOSH and MLS are you using?

Paragraph ending line 8, page 2: Substantial uncertainties also include those from artificial jumps introduced by changes in the observing system used as input to reanalyses. These jumps and their potential to impact temperatures that affect WV should be mentioned.

Page 5, line 28: Why does this analysis end in 2013? All of the reanalyses and obs extend to present, and cutting out 5 of the 15 years of the MLS record seems imprudent.

Section 2.3, page 6: This is more of a general comment, but this paper makes no mention of previous efforts at extracting SWV variability in obs and reanalysis-driven simulations. In particular, several papers by Dessler et al. have used a similar regression analysis. I believe the results of this paper are broadly consistent with the previous analyses, but some discussion of similarities and differences is warranted.

Dessler, A. E., Schoeberl, M. R., Wang, T., Davis, S. M. and Rosenlof, K. H.: Stratospheric water vapor feedback, PNAS, doi:10.1073/pnas.1310344110, 2013.

Dessler, A. E., Schoeberl, M. R., Wang, T., Davis, S. M., Rosenlof, K. H. and Vernier,

J. P.: Variations of stratospheric water vapor over the past three decades, J Geophys Res-Atmos, 119(22), 12,588–12,598, doi:10.1002/2014JD021712, 2014.

Page 6, line 32: "signals such as . . . QBO have zero long-term trend." This is not necessarily true over short periods of time where endpoint effects could come into play (e.g., if the record started in a westerly phase and ended in an easterly phase). This is not an issue for AC as long as full years of data are used (given that sine/cosine pairs are periodic by construction).

Page 7, line 7: ". . . regression model explains over 90% of the variations... " This is not correct. Fig 1 top panel shows an R=0.91, which means the percent variance explained is 0.91^2 = .83 -> 83%

Page 8, line 8: "Apprendix" -> "Appendix"

Page 8, line 25: Remove "a"

Page 10, line 5: You should cite Randel and Jensen here

Randel, W. J. and Jensen, E. J.: Physical processes in the tropical tropopause layer and their roles in a changing climate, Nature Publishing Group, 6(3), 169–176, doi:10.1038/ngeo1733, 2013.

Page 11, figure 4: Are the time periods used the same here between the reanalyses and obs?

Page 11, line 14: Should read "cold point that controls"

Page 13, line 10: I'm confused as to why the H2O_CH4 variations are opposite in phase relative to seasonal variations in H2Oe. I thought that the peak H2Oe (i.e., boreal summer) coincided with peak in-mixing, and hence anomalously high values of H2O_CH4 from old midlatitud air being mixed into the tropics. Please clarify.

Page 14, figure 6: What is month 0? Is it December? Or January?

Page 19, sentence beginning line 4: I think the consistency is more likely due to the greatly improved quality and quantity of SWV data from MLS, rather than a sudden improvement in the quality of the reanalyses. This possibility should at least be recognized in this paragraph.

Page 20, line 11 and line 18: Dessler et al. 2014 found similar results for volcano and ENSO impacts on SWV.

Page 20, line 31: "intercomparison" -> "intercompare"

Page 21, Table 1: Some measure of statistical significance would be useful here (e.g., put significant trends in bold)

Page 22, Figure 12, top row: This is the standard deviation, not the variance (i.e., the square of the standard deviation). The caption and text mistakenly refer to this as the variance.

Page 23, lines 12-14: This is hard to see given the coarse color scale in Figure 12.

Page 24, figure 13: It is hard to see the different symbols in this figure. This could be fixed by using a small horizontal offset of the symbols, or using symbols that can be more easily overlaid on one another.

Page 25, line 6: I don't think the phrase "in combination with the analysis increment" is needed in this sentence. It makes the sentence confusing.

Page 25, line 9: What is the "assimilation increment"? Maybe the authors mean the "assimilation tendency"?

Page 26, line 19: "produces more" -> "produces a more"

Page 26, line 28: "indicates" -> "indicate"

Page 27, line 8: text spacing messed up at the end of this line.

Page 28, line 12: remove "the"

---

## Referee Comment (RC3) · Anonymous Referee #3 · 18 Mar 2019

The paper presents an intercomparison of stratospheric water vapor as produced by the lagrangian transport model CLaMS driven by meteorological winds and temperature from three modern reanalyses. The results are compared with SWOOSH and MLS observations. In addition to comparing the magnitude of the entry values, which are dependent on the tropical tropopause temperature in each reanalysis, the annual cycle, QBO, ENSO and volcanic signals are compared, as well as the linear trends. The results are accurately presented and the paper is well written. I recommend publication after the following minor issues are corrected.

- Page 1 Line 15: cloud effect

[Figure]

- Page 6 Line 19: here and in other parts of the paper: the QBO is not a periodic signal, please correct 'periodic' by 'quasi-periodic'.

- Section 2.3: The authors are confusing the SAO signal, which is present in the upper stratosphere, with a semi-annual harmonic component of the annual cycle. The terminology 'SAO' should only refer to the former.

- Page 7 Line 7: 'variation' should be 'variance'

- Page 11 Line 8: 'tropical tropopause temperatures': are these Lagrangian cold point temperatures? Please specify.

- Page 12 Line 17: In addition to tropical upwelling, Glanville and Birner (2017, ACP) argue that mixing effects could be important for the tape recorder. This relevant information could be included here, as it implies that not all differences in the tape recorder signal should be attributed to tropical upwelling.

- Page 14 Line 12: 'Although we use different methods to estimate the AC amplitude': Why are different methods used and which method is used here?

- Page 15 Line 5-6: can you point to specific 'small-scale processes that must be parameterized in the model'?

- Figures 8 and 10: I do not understand the meaning of the arrows, please explain more clearly.

- Page 16 Line 9: quasi-periodic

- Section 5 (Figure 10): it would be much easier for the reader if you describe the interpretation of the QBO phase representation in Fig. 10 here, instead of having to go back and look for the information.

- Figure 11 caption: Please remind the reader that these are values at 400 K.

- Page 20 Line 21: Why is the lag for the ENSO signal on H2O entry anomalies so

long? One year seems an excessive time lag, since the signal in tropical upwelling maximizes only after a few months.

- Page 21 Line 5: since this result is not shown in Table I, I recommend adding '(not shown)'

- Page 25 Line 33: remove 'relatively'

- Page 27 Line 8: typo 'n ext'
* * *

---

## Author Comment (AC1) · 16 Apr 2019

Reply to Referee #1

This paper uses a single model (CLaMS) to calculate stratospheric water vapour values on the basis of a simple dehydration scheme combined with a simple methane oxidation scheme. The model is driven by reanalysis winds and temperatures (winds to determine transport and temperature to specify dehydration). Three different reanalysis datasets (ERA-I, JRA-55 and MERRA-2) are used to drive the model and the objective of the paper to compare predictions of stratospheric water vapour concentrations that result from the use of each of the three datasets. I think that this paper potentially makes a useful contribution. Reanalysis datasets are in wide use for many different types of calculations and it is important to have on record different measures of the differences between the datasets in common use. There have been previous studies of the differences in transport characteristics between these datasets, but the calculation of stratospheric water vapour considered here on the one hand requires a particular combination of transport and temperature information and on the other is of significant general interest, because of the radiative and chemical importance of stratospheric water vapour and of the continuing challenges in measuring its concentrations and in establishing a long-term observational record.

*We are grateful for the reviewer1's comments and suggestions, which are very helpful to improve this work. The changes corresponding to the comments can be tracked by the text in red in the manuscript. The responses or clarifications are specified following each suggestion below.*

However, I do consider that the paper could be improved in various ways to make it as valuable as possible to researchers working in this general subject area.

1. p2 l21: 'water vapour values entering the stratosphere are determined by ... in the upper troposphere' - seems odd statement - previous papers (e.g. Liu et al 2010, Fig. 12) have suggested LDPs are distributed over a layer centred on about 90hPa. 'in the TTL' would be a more usual description.

   *Agree, the sentence has been reformulated.*

2.  p2 l22: 'Lagrangian models provide more accurate records of the temperature histories of air parcels compared to Eulerian models' – statement is true by definition (Eulerian models don't provide such records) or could be misread as 'provide a more accurate representation of transport' which is questionable - there are advantages and disadvantages to the two different formulations.

    *Agreed, the sentence has been reformulated as 'Lagrangian approaches provide accurate records of the temperature histories of air parcels that Eulerian models cannot provide, and therefore provide more reliable representations of entry mixing ratios in SWV'.*

3.  p3 l13-: '30% too fast', '30% too slow' are based on the tape recorder considered over some a particular range of heights - give range explicitly.

    *The range has been added. The altitude given in Schoeberl et al. (2012) is 17-22km.*

4.  p4 l8: You don't give sufficient information here to specify the dehydration scheme – e.g. the fall speed has to be combined with a length scale (specified in Ploeger et al 2013 as 300m). A first point is that if I look at the references you give – e.g. Poshyvailo et al (2018) I don't see sufficient information to be able to reproduce the scheme (e.g. that paper doesn't seem to say what the assumed fall speed it). My suggestion is that you provide all the necessary information together in an Appendix. A second point is that you mention various details such as sedimentation rate but then you fail to mention the length scale - which is surely chosen on a completely ad hoc basis. So the naive reader might will infer that dehydration scheme is based on a precise physical model, whereas in fact it requires choice of length scale - which is surely ad hoc. (Ploeger et al 2013 note that it is comparable to the vertical resolution of the model - but that is hardly a physical justification.) So more clarity is needed - both for reproducibility and for an honest description of what is being done.

    *Here we added the information: 'The sedimentation fall speed is calculated by assuming a mean ice particle radius. Ice sedimentation is then determined by the comparison of the sedimentation length over the model time step against a characteristic length ~300m (Hobe et al., 2011; Ploeger et al., 2013).'*

5. p4 l12: The information that greater intensity of small-scale mixing leads to moistening by 0.5ppmv and amplification of the annual cycle by 0.2 ppmv can't really be interpreted sensibly without more information. For example you could say that the range of mixing strength being considered is representative of actual uncertainty in small-scale mixing (which is what is said in the Poshyvailo et al 2018 paper).

*Agree. The statement has been changed to 'the range of mixing strength they considered was representative of actual uncertainty in small-scale mixing'.*

6. p4 l14: 'best agreement with MLS observations' - I found this sentence a bit misleading. Of the papers cited, only Poshyvailo et al 2018 gives explicit information on the effect of varying mixing strength. So I don't see that the others provide any useful information on what choice of mixing strength gives the best agreement with MLS.

*It is true. The other two studies used long-term CLaMS water vapor output which is valid with MLS observations. However, that is not the sensitivity study of mixing strength. Thus, they are removed.*

7. p4 l23: How are you calculating saturation mixing ratio exactly? Simmons et al (1999 QJRMS) make the point that the precise form of the expression used for saturation mixing ratio can be important. Again this is the sort of information that needs to be easily available to ensure that others can reproduce your calculations.

Saturation mixing ratio is calculated by:

$$\chi_{H2O} = p_s/p_{cp}$$

where $p_s$ is given by $10^{-2663.5/T+12.537}$ (Marti and Mauersberger, 1993).

The information has been added to manuscript in Page 4 Line 6-7.

8. p4 l25: 'The effects of tuning ...' - this sentence would be clearer as 'The effects of tuning the critical supersaturation threshold in CLaMS have a similar effect to the effect of applying a frost point offset to the Lagrangian dry point temperature noted by Liu et al., (2010) and Fueglistaler et al., (2013), in that increase in the

supersaturation threshold enhances both the mean value and the amplitude of the annual cycle in simulated H2O.'

*We reformulated the sentence as the reviewer recommended.*

9. p4 l28: I couldn't follow the logic of these two sentences - to me the point is that given uncertainty over the precise relation between temperature and saturation, e.g. due to the uncertainty in the appropriate value of supersaturation threshold, it is impossible to be interpret differences between predicted water vapour and observations as being due to errors in LDP temperatures.

*The sentence is revised to 'Due to the uncertainty in the appropriate value of supersaturation thresholds, the differences among the modelled values of $H_2O$ entry mixing ratio or between modelled values against observations cannot be unequivocally interpreted as errors in Lagrangian dry point temperatures in these reanalyses'.*

10. p4 l16: My experience is that the re-analyses don't provide diabatic heating as a single quantity, but as various components. Confirm that you are using all components (e.g. including latent heating).

*We deduce cross-isentropic vertical velocity from the total diabatic heating rates, which includes the all-sky radiation, latent heat release as well as the diffusive turbulent heat transport from each reanalysis product. The information has been added to the manuscript.*

11. p5 l5: My understanding is that you are estimating methane supped water from (1), using a mode prediction of CH_4^rec. But how are you specifying alpha?

*It was not so clear in the previous version. Alpha is calculated following $CH_4^{rec}$ (eq. 2), by $(CH_4^{rec}-CH_4)/CH_4^{rec}$. This equation is now isolated from eq.(1).*

12. p6 l12-15: It seems important not to ignore the fact that the MLS weighting functions do not only produce 'artefacts' when applied to the CLaMS output, but that these artefacts may also be part of what is presented as the MLS observation - i.e. the MLS observations imply vertical structure that may be quite

different to what is actually present. (At least that was my interpretation of Ploeger et al 2013.)

*It is true. We compared the differences between CLaMS results of AC amplitude with and without application of AK (see the figure below). The differences are only visible in high latitudes at lower stratosphere. Similar result is shown in Ploeger et al 2013.*

[Figure]

*The region showing differences (high latitudes at lower stratosphere) are not the main concern in this paper.*

*We made a note at Page 6 line 23-24.*

13. p6 l20: My understanding here is that you are simply extracting the semiannual cycle on the basis of its semi-annual frequency - not implying a direct relation to the phenomenon 'the Semi-Annual Oscillation (SAO)' that is identified in the low-latitude upper stratosphere. In other words, there is a potential difference in meaning between 'the SAO' and 'the semi-annual harmonic'.

*Yes, we confused SAO with semi-annual harmonic. The related terms are revised to semi-annual harmonic throughout the manuscript.*

14. p6 l26: It would be helpful here to note that the amplitudes and phases are what is presented in specific later figures.

*The corresponding figures are specified.*

15. p8 l21: It's a minor point, but you don't seem to have said explicitly that CLaMS-MRA is the MERRA simulation.

*The brief name of three CLaMS runs is explained in the new manuscript at Page 5 line 17-18.*

16. p12 l7: 'Moreover, this makes it possible that an optimisation of the dehydration scheme...' - this sentence doesn't seem very relevant at this point in the text (and you have said something similar on p4).

*Agree, this repeated information is deleted.*

17. p12 l14: 'However the AC phase of $H_2O$ entry values based on CLaMS-MRA is in better agreement with the SWOOSH data' - given the uncertainty indicated in SWOOSH one wonders whether the better agreement is actually significant. The comment is mostly about September/October - i.e. the judgement all comes down to September-October differences. Whether or not CLaMS-MRA or ClaMS-JRA is in better agreement with SWOOSH during the J-F-M-A period is also open to question.

*The fact is that it is hard to judge which run shows the best phase agreement to SWOOSH. For the boreal spring when the minimum comes, the shapes of three runs are quite close. The visible differences only show for the maximum months. However, we agree that the differences are not significant. Thus, here we only state the fact and we also reformulated this part in conclusion accordingly.*

18. p12 l31: For me a strong part of the evidence for the importance of different regions are LDP distributions based on trajectory studies - again Fueglistaler et al (2005) given the relevant distributions.

*The citation has been added.*

19. p13 l6: Can the slow upwelling alone account for the seasonal signal in $H2O\_CH\_4$? - i.e. isn't the seasonal cycle in in-mixing at least as important?

*Yes, we reformulated to 'This feature is a joint effect of the slow tropical upwelling and stronger in-mixing from the extratropics, resulting in...'.*

20. p14 Figure 6: It would be helpful to have the propagation of the maximum values as well as the propagation of the minimum values explicitly marked on the Figures.

*We did not plot both the maximum and minimum propagation lines. The first reason is that the propagation line now is determined by the largest correlation of each layer with the layer below. Therefore, this propagation should represent both the maximum and minimum propagations. The second reason is to avoid the complication in the figure since it already has two propagation lines in each panel: one from the CLaMS run and one from SWOOSH for comparison. If we follow this idea, there will be four lines in each panel without giving more information.*

21. p15/16: Figures 7 and 8 - it would be much better if Figure 7 could be immediately followed by Figure 8 - perhaps they could even be combined into a single Figure. I found the overlaid dots in Figure 7 quite difficult to see and to interpret – of course when reading the paper on a screen one can zoom to look in more detail at interesting features, so one isn't applying the same rules of legibility that might have been applied for paper publications. But given that the dots represent relative error, the fact that some of them are the same colours as are used for the values of the depicted field is confusing. It might be clearer to do something like use overprinted + and - (of different size to indicate size of relative difference).

*We agree. Here we combined the two figures (also later figures of QBO). And the overlaid dots are removed for Figure 7, which are then replaced by symbols '+' and '-'. The plus and minus symbols are shown in a schematic way, where is discussed in the text. Although some accurate information about the differences are lost, the figure becomes more reader-friendly.*

22. p17 l3: 'The phase of the QBO effect at the tropopause is therefore consistent with that of the 50hPa QBO wind' - the 'therefore' only makes sense if you say explicitly that it is the 50hPa wind (rather than the wind at some other level) that is correlated with the tropopause temperature.

*Since the tropopause is colder (warmer) during the easterly (westerly) phase of the QBO (referring to 50hPa wind) (Plumb and Bell, 1982), the phase of the QBO effect at the tropical tropopause shown in Fig.8 is related to the phase of 50hPa QBO wind. The change is made in Page 17 line 13-15.*

23. p17 l6: 'both overestimate A_QBO at isentropic levels between 450K and 550K' ... 'both biases can be traced back to strong diabatic upwelling' - the key point surely is not necessarily that the diabatic upwelling is strong, but that the the circulation (upwelling + eddy mixing) is such that the tape recorder signal in these two analyses decreases in amplitude in the vertical slowly relative to MLS and MERRA (as you show for the annual cycle in Figure 6 and 7).
*We agree that both upwelling and eddy mixing (the circulation) are working. Moreover, slow upwelling makes the residence time longer, which will give more time for mixing process (at least in the model) and thus bring more exchange between tropics and extratropics. The sentence has been changed, please find it at Page 17 line 22-23.*

24. p17 Figure 9: Give information in the caption on the dashed box. You could call this 'region 3'. Incidentally this region doesn't really seem to be characterised by a 'clear peak' - contrary to what you say on p16 l14.
*It is named to region 3 as suggested. And we changed the statement about 'clear peak' as 'A peak in the tropical middle stratosphere (region 3) is clearly shown in CLaMS-ERA and CLaMS-JRA but not clear in MLS and CLaMS-MRA.'*

25. p17 l9: You say 'the large values of A_QBO in region 2 are mainly linked to QBO related modulation of the stratospheric circulation.' You should provide a reference for this. For example this aspect of the QBO signal was considered in the Baldwin et al (2001) review. That was quoting previously published studies. There may have been more recent work on this topic.
*The citation is added.*

26. p17 l11: This reference should be 'Lossow et al 2017b'. The 2017a reference doesn't mention the QBO at all.

*It is corrected.*

27. p18 l3: 'The amplitude and phase of the QBO signal in SWV show pronounced uncertainty in the middle stratosphere' - what do you mean by this? Do you mean that the observational signal is uncertain? (I don't think so.) If you mean that there is disagreement between the different CLaMS simulations and between the CLaMS simulations and the re-analysis then say that directly.
*Yes, we meant disagreement between the different CLaMS simulations. The sentence is rewritten.*

28. p18 Figure 10: As previously noted for Figures 7 and 8, it would be helpful for the reader if Figures 9 and 10 were immediately adjacent (or combined into a single Figure).
*The original Fig. 9 and 10 are combined as new Figure 8 and we revised as we did for original Fig. 7 and 8.*

29. p18 A key point about the QBO signal in water vapour is that it results in part from upward propagation of the water vapour variations 'imprinted' by the QBO temperature variation at the tropical tropopause and in part from QBO variations in the meridional circulation (which seem likely to be primarily to be responsible for the QBO signal in water vapour seen in the upper stratosphere). You don't really exploit the distinction between these two processes generally in interpreting Figure 10 (which at first sight looks rather complicated). For example, you have identified from the annual cycle that ERA-I and JRA-55 seem to propagate signals too rapidly in the vertical in the lower stratosphere – presumably this means that their QBO signals are different in phase to the observed in the lower stratosphere, though you don't identify this as a significant difference (i.e. there is no vertical arrow marked corresponding to this in the lower stratosphere), presumably because the associated phase error is only a month or so. You do mark an arrow for JRA-55 - but don't comment on it. Do you interpret this as resulting from the strong vertical upwelling in JRA-55?

*After checking the phase difference again for an interval of 1 month, we confirm that differences in the region1 is around 1 month. Therefore, we added the smaller arrows indicate 1 month differences.*

*In the revised figure 8, it is seen that the phase of CLaMS-ERA and CLaMS-JRA in region 1 again propagate faster with larger amplitudes. CLaMS-MRA shows the contradictious result for the MLS period and whole period, which seem to result from the influence of QBO-wind differences before and after 1990 to FUB wind as shown also in region 3.*

*The text is revised accordingly.*

30. p18 l12: The fact that MERRA-2 doesn't match the observed QBO during the 1980s and early 1990s seems a pretty major and obvious problem to me. If that is the main cause of the difference in phase errors for MERRA between Figures D1 and D2 (indicated by the arrows in different directions) then it is probably clearer to omit D2 (and say why you have done that).

*Agree, we follow the suggestion.*

31. p26 l6: I've noted previously that the difference in annual cycle phase between predictions of different reanalyses seems to be judged almost on the basis of small differences between two months - and my question is whether that difference is really significant.

*We agreed that the difference is not significant. As we mentioned in the response to question 17, this part of text has been revised in a similar way.*

---

## Author Comment (AC2) · 16 Apr 2019

Reply to Referee #2

This paper evaluates the simulated stratospheric water vapor in three modern reanalyses in comparison with observations to better understand the inter-reanalysis differences.

Overall, this paper is well written and well-supported in its arguments, and will be a nice contribution to the literature particularly for those wishing to better understand how the choice of input reanalysis for models (such as CTMs) will impact model results.

*We are grateful for reviewer2's suggestions, which are very helpful to improve the paper. We mark the changes in the manuscript in blue color. The responses or clarifications are specified following each suggestion below.*

I have a few minor suggestions that should be considered:

1. Title/abstract: I believe through most of the paper the British English spelling of vapor (i.e., vapour) is used, but the American English spelling is used in the title and abstract. A consistent spelling should be used throughout the paper.

   *All has been revised to 'water vapor' consistently.*

2. Line 17: There have been other attempts at merging observational records of WV. In particular, Froidevaux et al 2015.

   *Yes, it is added as citation.*

3. Lines 30-31: What versions of SWOOSH and MLS are you using?

   *We use SWOOSH version2.5 and MLS version 4.2, which has been included in corresponding places in the manuscript.*

4. Paragraph ending line 8, page 2: Substantial uncertainties also include those from artificial jumps introduced by changes in the observing system used as input to reanalyses. These jumps and their potential to impact temperatures that affect WV should be mentioned.

   This information is added in the Page2 line 9-10.

5. Page 5, line 28: Why does this analysis end in 2013? All of the reanalyses and obs. extend to present, and cutting out 5 of the 15 years of the MLS record seems imprudent.

*The period of 1979.01-2013.12 is noted as the "S-RIP base period", which makes the inter-comparison easier among various studies.*

6. Section 2.3, page 6: This is more of a general comment, but this paper makes no mention of previous efforts at extracting SWV variability in obs. and reanalysis-driven simulations. In particular, several papers by Dessler et al. have used a similar regression analysis. I believe the results of this paper are broadly consistent with the previous analyses, but some discussion of similarities and differences is warranted.

*Yes, we agree that the studies by Dessler et al. should be mentioned, especially in section 6.*

*The changes for this point is added at Page19 line 21-25.*

7. Page 6, line 32: "signals such as . . . QBO have zero long-term trend." This is not necessarily true over short periods of time where endpoint effects could come into play (e.g., if the record started in a westerly phase and ended in an easterly phase). This is not an issue for AC as long as full years of data are used (given that sine/cosine pairs are periodic by construction).

*Agreed. More precisely, the QBO might contribute to trend even when the full cycle is taken. So information is added as 'The quasiperiodic signal like QBO does not show long-term trend in the $H_2O$ entry over the considered period' at Page 7 line 10-11.*

,

8. Page 7, line 7: ". . . regression model explains over 90% of the variations... " This is not correct. Fig 1 top panel shows an R=0.91, which means the percent variance explained is 0.91^2 = .83 -> 83%

*Thanks for pointing it out, it is revised.*

9. Page 8, line 8: "Apprendix" -> "Appendix"

*Corrected.*

10. Page 8, line 25: Remove "a"

*Corrected.*

11. Page 10, line 5: You should cite Randel and Jensen (2013) here

*Agreed, the citation has been added.*

12. Page 11, figure 4: Are the time periods used the same here between the reanalyses and obs.?

*No, we used the full simulation period (1980.01-2013.12) for CLaMS runs, which is longer than the SWOOSH period. We clarify this in the caption of Figure 4. The main objective of this figure is to compare the climatological AC among reanalysis while the validation with SWOOSH data is only supporting information. Thus, we did use the same period. For your information, the conclusion stays the same when we use exactly the same 'SWOOSH period' for model results, with only some quantity changes. Please check the plot below when using 'SWOOSH period' for all.*

[Figure]

13. Page 11, line 14: Should read "cold point that controls"

*Corrected.*

14. Page 13, line 10: I'm confused as to why the H2O_CH4 variations are opposite in phase relative to seasonal variations in H2Oe. I thought that the peak H2Oe (i.e., boreal summer) coincided with peak in-mixing, and hence anomalously high values of H2O_CH4 from old midlatitud air being mixed into the tropics. Please clarify.

*It is true that the $H2O_{CH4}$ peak coincides with the peak of H2Oe. But these two signals occur at different altitudes. H2Oe seasonal cycle starts at the tropical tropopause while $H2O_{CH4}$ seasonal cycle due to in-mixing of mid-latitude air occurs over the full layer from 380-450 K (in CLaMS-MRA with slow circulation in the lower stratosphere). Therefore, they have a shift of phases when these two seasonal signals (or 'tape recorder') propagates to the same altitude.*

15. Page 14, figure 6: What is month 0? Is it December? Or January?

*Month 0 is January and month 12 is Jan. again. We didn't notice it is a bit odd to label x-axis in this way. The labels have been changed explicitly to the month names.*

16. Page 19, sentence beginning line 4: I think the consistency is more likely due to the greatly improved quality and quantity of SWV data from MLS, rather than a sudden improvement in the quality of the reanalyses. This possibility should at least be recognized in this paragraph.

*Agreed. There are two aspects: 1) the improvement of SWOOSH data due to the kick-in of MLS; 2) the improvement in the quality of the reanalyses. The sentences have been reformulated.*

17. Page 20, line 11 and line 18: Dessler et al. 2014 found similar results for volcano and ENSO impacts on SWV.

*Added.*

18. Page 20, line 31: "intercomparison" -> "intercompare"

*Corrected.*

19. Page 21, Table 1: Some measure of statistical significance would be useful here (e.g., put significant trends in bold)

*The significant trends in the table have been marked in bold.*

20. Page 22, Figure 12, top row: This is the standard deviation, not the variance (i.e., the square of the standard deviation). The caption and text mistakenly refer to this as the variance.

*The caption of figure and the relevant text are revised.*

21. Page 23, lines 12-14: This is hard to see given the coarse color scale in Figure 12.

*Yes, the sentence is too strong. Actually, the following arguments are just intended to say the variance fractions are 'qualitatively' consistent. Therefore, we revised this sentence accordingly.*

22. Page 24, figure 13: It is hard to see the different symbols in this figure. This could be fixed by using a small horizontal offset of the symbols, or using symbols that can be more easily overlaid on one another.

*The smaller symbols in each panel are changed to another shape, which makes them easier to be seen. However, we insist to plot the symbols together. In this way, it emphasizes virtually that the same bins are used for each reanalysis. Meanwhile, since the symbols are repeated in each panel, the offset of the symbols seems not so necessary.*

23. Page 25, line 6: I don't think the phrase "in combination with the analysis increment" is needed in this sentence. It makes the sentence confusing.

*Changed as the suggestion.*

24. Page 25, line 9: What is the "assimilation increment"? Maybe the authors mean the "assimilation tendency"?

*Yes, we meant "analysis tendency". Changed as suggestion.*

25. Page 26, line 19: "produces more" -> "produces a more"

    *Corrected.*

26. Page 26, line 28: "indicates" -> "indicate"

    *Corrected.*

27. Page 27, line 8: text spacing messed up at the end of this line.

    *Corrected.*

28. Page 28, line 12: remove "the"

    *Corrected.*

---

## Author Comment (AC3) · 16 Apr 2019

Reply to reviewer 3

The paper presents an intercomparison of stratospheric water vapor as produced by the Lagrangian transport model CLaMS driven by meteorological winds and temperature from three modern reanalyses. The results are compared with SWOOSH and MLS observations. In addition to comparing the magnitude of the entry values, which are dependent on the tropical tropopause temperature in each reanalysis, the annual cycle, QBO, ENSO and volcanic signals are compared, as well as the linear trends. The results are accurately presented and the paper is well written. I recommend publication after the following minor issues are corrected.

*We are grateful for reviewer3's suggestions, which are very helpful to improve this work. The changes corresponding to the comments can be tracked by the text in orange in the manuscript. The responses or clarifications are specified following each suggestion below.*

- Page 1 Line 15: cloud effect

*Corrected to 'long-wave cloud radiative effect'.*

'

- Page 6 Line 19: here and in other parts of the paper: the QBO is not a periodic signal, please correct 'periodic' by 'quasi-periodic'.

*Corrected throughout the paper.*

- Section 2.3: The authors are confusing the SAO signal, which is present in the upper stratosphere, with a semi-annual harmonic component of the annual cycle. The terminology 'SAO' should only refer to the former.

*We agree. This point was also made by Reviewer 1. The usages of 'SAO' are corrected to semi-annual harmonic (SAH).*

- Page 7 Line 7: 'variation' should be 'variance'

*Corrected.*

- Page 11 Line 8: 'tropical tropopause temperatures': are these Lagrangian cold point temperatures? Please specify.

*No, they are not the Lagrangian cold point. Here tropical tropopause temperatures refer to the minimum of tropical mean (averaged over 20S-20N) temperature between $\theta=360K$ and 420K. The caption is revised accordingly.*

- Page 12 Line 17: In addition to tropical upwelling, Glanville and Birner (2017, ACP) argue that mixing effects could be important for the tape recorder. This relevant information could be included here, as it implies that not all differences in the tape recorder signal should be attributed to tropical upwelling.

*Thanks for the remark, which is indeed relevant. The information is added with the reference at Page 13 line 2-3.*

- Page 14 Line 12: 'Although we use different methods to estimate the AC amplitude': Why are different methods used and which method is used here?

*In section 4.1, we simply used climatological monthly mean to represent AC as shown in Figure 6. The phase propagation is determined by strongest correlation for each layer to the layer below. In section 4.2 (original Fig. 7 and 8; new Fig.7 in the revised version), the harmonic regression with a sine and cosine term is used to determine the amplitude and phase of AC.*

- Page 15 Line 5-6: can you point to specific 'small-scale processes that must be parameterized in the model'?

*More explicitly, that means the choice of mixing parameter (or mixing strength) in the model (which describes the small-scale processes in reality) might be of importance in the modelling of water vapor in the Southern Hemisphere subtropical lower stratosphere.*

- Figures 8 and 10: I do not understand the meaning of the arrows, please explain more clearly.

*The arrows show the primary regions where the phase differs from the benchmark phase (referring to the leftmost panel in each row). Upward arrows show phases that*

*lag behind the benchmark phase while downward arrows show phases that lead the benchmark phase.*

- Page 16 Line 9: quasi-periodic

*Corrected.*

- Section 5 (Figure 10): it would be much easier for the reader if you describe the interpretation of the QBO phase representation in Fig. 10 here, instead of having to go back and look for the information.

*Agree. We now combined the two figures as the new Figure 8 for QBO and did the same for AC (new figure 7).*

- Figure 11 caption: Please remind the reader that these are values at 400 K.

*The information is added to the caption of Figure 9 in revised manuscript.*

- Page 20 Line 21: Why is the lag for the ENSO signal on H2O entry anomalies so long? One year seems an excessive time lag, since the signal in tropical upwelling maximizes only after a few months.

*The lag for ENSO from our regression is 10-11 months. There are two reasons for the long lag time:*

1) *The lag time is location-dependent. When considering the zonally resolved lag time distribution for AoA at 17km to ENSO, the response time is mostly below 4 months. However, the tropical mean data we used shows lag time around 10-11 months. One potential reason is the effect of smoothing of localized signals.*

2) *Note that we used H2O at 400 K isentrope. The propagation from cold point tropopause to 400K typically takes 1-2 months.*

- Page 21 Line 5: since this result is not shown in Table I, I recommend adding '(not shown)'

*Added.*

- Page 25 Line 33: remove 'relatively'

*Removed.*

- Page 27 Line 8: typo 'n ext'

*Corrected.*

---

## Editor Decision (ED1)

**Technical corrections on Manuscript No ACP-2019-39:**

P3, L15: add space between number and unit

P3, L15: temove space between "layer" and full stop.

P3, L16: remove space between "range" and comma.

P3, L21: remove space between paranthesis and "which".

P4, L5: details → detail

P4, L7: remove space between "tracjectory" and comma.

P4, L34: Modelled → modelled

P7, L9: eq. → Eq.

P7, L11: H2O → subscript is missing.

P7, L14: sect. 6 → Sect. 6

P7, L24: eq. → Eq.

P7, L26: section 4, section 5 and section 6 → Sect.4, Sect 5. and Sect. 6
however, better would be to write directily Sect. 4 to 6.

P8, L3: eq. (5) → Eq. (5)

P9, , L4: 192.5 ? Is that correct?

P10, L2: space between Figure 2 and text afterwards missing?

P15, L16 and L21: section 4.1 → Sect. 4.1

P17, L8: Figure 8 → Fig. 8

P17, L30: add space between Fig. 2 and (A3) or write Fig.2A3
Here, I would recommend you to check the ACP guidelines how to do the correct sublabeling of the
fugures.

P19, L21: remove space between "model" and comma.

P19, L23: regreesor → regressor

P23, Fig 10 caption: remove space between paranthesis and sigma.

P26, L17: "a moderate H2O entry mixing ratios" → either use singular or plural.

P27, L6: section4.1 → Sect. 4.1

P27, L22: section 2.2 → Sect. 2.2

---

## Author Response (AR2)

All the corrections are done.

We checked over the manuscript for all 'section *', and changed them to 'Sect. *'.

P8, L3: eq. (5) → Eq. (5) P9, , L4: 192.5 ? Is that correct?

Yes, it is 192.5